# Anytime-Competitive Reinforcement Learning with Policy Prior

**Jianyi Yang**
UC Riverside
Riverside, CA, USA
jyang239@ucr.edu

**Pengfei Li**
UC Riverside
Riverside, CA, USA
pli081@ucr.edu

**Tongxin Li**
CUHK Shenzhen
Shenzhen, Guangdong, China
litongxin@cuhk.edu.cn

**Adam Wierman**
Caltech
Pasadena, CA, USA
adamw@caltech.edu

**Shaolei Ren**
UC Riverside
Riverside, CA, USA
shaolei@ucr.edu

## Abstract

This paper studies the problem of Anytime-Competitive Markov Decision Process (A-CMDP). Existing works on Constrained Markov Decision Processes (CMDPs) aim to optimize the expected reward while constraining the *expected* cost over random dynamics, but the cost in a specific episode can still be unsatisfactorily high. In contrast, the goal of A-CMDP is to optimize the expected reward while guaranteeing a bounded cost in *each* round of *any* episode against a policy prior. We propose a new algorithm, called Anytime-Competitive Reinforcement Learning (`ACRL`), which provably guarantees the anytime cost constraints. The regret analysis shows the policy asymptotically matches the optimal reward achievable under the anytime competitive constraints. Experiments on the application of carbon-intelligent computing verify the reward performance and cost constraint guarantee of `ACRL`.

## 1 Introduction

In mission-critical online decision-making problems such as cloud workload scheduling [49, 20], cooling control in datacenters [63, 16, 46], battery management for Electrical Vehicle (EV) charging [57, 36], and voltage control in smart grids [54, 73], there is always a need to improve the reward performances while meeting the requirements for some important cost metrics. In these mission-critical systems, there always exist some policy priors that meet the critical cost requirements, but they may not perform well in terms of the rewards. In the application of cooling control, for example, some rule-based heuristics [19, 46] have been programmed into the real systems for a long time and have verified performance in maintaining a safe temperature range, but they may not achieve a high energy efficiency. In this paper, we design a Reinforcement Learning (RL) algorithm with the goal of optimizing the reward performance under the guarantee of cost constraints against a policy prior for any round in any episode.

Constrained RL algorithms have been designed to solve various Constrained MDPs (CMDPs) with reward objective and cost constraints. Among them, some are designed to guarantee the expected cost constraints [61, 62], some can guarantee the cost constraints with a high probability (w.h.p.) [13], and others guarantee a bounded violation of the cost constraints [26, 29, 21, 22, 1]. In addition, conservative RL algorithms [27, 68, 31, 65] consider constraints that require the performance of a learning agent is no worse than a known baseline policy in expectation or with a high probability. [27] points out that it is impossible to guarantee the constraints for any round with probability

37th Conference on Neural Information Processing Systems (NeurIPS 2023).

| Methods | Unknown dynamic | Expected constraints or constraints w.h.p. | | Any-episode constraints | Anytime-competitive constraints |
|---|---|---|---|---|---|
| | | With violation | No violation | | |
| Learning-augmentation | ✗ | N/A | N/A | [38, 51, 17] | ✗ |
| Constrained RL | ✔ | [29, 21, 22, 1, 67, 26] | [62, 61] | ✗ | ✗ |
| Conservative RL | ✔ | N/A | [68, 27, 31, 65] | ✗ | ✗ |
| **ACRL (this work)** | ✔ | N/A | ✔ | ✔ | ✔ |

Table 1: Comparison between ACRL and most related works.

one while such guarantees are desirable. In real mission-critical applications, the cost constraints are often required to be satisfied at each round in any episode even in the worst case, but such anytime constraints are not guaranteed by the existing constrained/conservative RL policies. Recently, learning-augmented online control algorithms [37, 38, 69, 51, 17, 35] have been developed to exploit machine learning predictions with the worst-case control performance guarantee. Nonetheless, the learning-augmented control algorithms require the full knowledge of the dynamic models, which limits their applications in many systems with unknown random dynamic models. A summary of most relevant works is given in Table 1.

To fill in this technical blank, we model the mission-critical decision-making problem as a new Markov Decision Process (MDP) which is called the Anytime-Competitive MDP (A-CMDP). In A-CMDP, the environment feeds back a reward and a cost corresponding to the selected action at each round. The next state is updated based on a random dynamic model which is a function of the current action and state and is not known to the agent. The distribution of the dynamic model is also unknown to the agent and needs to be learned. Importantly, at each round $h$ in any episode, the policy of A-CMDP must guarantee that the cumulative cost $J_h$ is upper bounded by a scaled cumulative cost of the policy prior $\pi^\dagger$ plus a relaxation, i.e. $J_h \leq (1 + \lambda)J_h^\dagger + hb$ with $\lambda, b > 0$, which is called an *anytime* competitive constraint or *anytime* competitiveness. The term "competitive" or "competitiveness" is used to signify the performance comparison with a policy prior. Under the anytime cost competitiveness for all rounds, the RL agent explores the policy to optimize the expected reward.

The anytime competitiveness guarantee is more strict than the constraints in typical constrained or conservative MDPs, which presents new challenges for the RL algorithm design. First of all, the anytime competitive constraints are required to be satisfied for any episode, even for the early episodes when the collected sequence samples are not enough. Also, to guarantee the constraints for each round, we need to design a safe action set for each round to ensure that feasible actions exist to meet the constraints in subsequent rounds. Last but not least, without knowing the full transition model, the agent has no access to the action sets defined by the anytime competitive constraints. Thus, in comparison to the control settings with known transition models [38], ensuring the anytime competitiveness for MDPs is more challenging.

**Contributions**. In this paper, we design algorithms to solve the novel problem of A-CMDP. The contributions are summarized as follows. First, we propose an Anytime-Competitive Decision-making (ACD) algorithm to provably guarantee the anytime competitive constraints for each episode. The key design in ACD is a projection to a safe action set in each round. The safe action set is updated at each round according to a designed rule to gain as much flexibility as possible to optimize the reward. Then, we develop a new model-based RL algorithm (ACRL) to learn the optimal ML model used in ACD. The proposed model-based RL can effectively improve the reward performance based on the new dynamic defined by ACD. Last but not least, we give rigorous analysis on the reward regret of ACRL compared with the optimal-unconstrained policy. The analysis shows that the learned policy performs as well as the optimal ACD policy and there exists a fundamental trade-off between the optimization of the average reward and the satisfaction of the anytime competitive constraints.

## 2 Related Work

**Constrained RL**. Compared with the existing literature on constrained RL [68, 1, 67, 10, 13, 2, 21, 29, 26, 22, 61], our study has important differences. Concretely, the existing constrained RL works consider an *average* constraint with or without constraint violation. In addition, existing conservative RL works [68] consider an *average* constraint compared with a policy prior. However, the constraints can be violated especially at early exploration episodes. In sharp contrast, our anytime competitive

constraint ensures a strict constraint for any round in each episode, which has not been studied in the existing literature as shown in Table 1. In fact, with the same policy prior, our anytime competitive policy can also meet the average constraint without violation in conservative/constrained RL [68, 25]. [56] considers MDPs that satisfy safety constraint with probability one and proposes an approach with a high empirical performance. However, there is no theoretical guarantee for constraint satisfaction. Comparably, our method satisfies the constraint with theoretical guarantee, which is essential to deploy AI for mission-critical applications.

Our study is also relevant to safe RL. Some studies on safe RL [10, 2, 13] focus on constraining that the system state or action at each time $h$ cannot fall into certain pre-determined restricted regions (often with a high probability), which is orthogonal to our anytime competitiveness requirement that constrains the cumulative cost at each round of an episode. Our study is related to RL with safety constraints [10, 39], but is highlighted by the strict constraint guarantee for each round in each episode. In a study on safe RL [10], the number of safety violation events is constrained almost surely by a budget given in advance, but the safety violation value can still be unbounded. By contrast, our work strictly guarantees the anytime competitive constraints by designing the safety action sets. In a recent study [3], the safety requirement is formulated as the single-round cost constraints. Differently, we consider cumulative cost constraints which have direct motivation from mission-critical applications.

**Learning-augmented online decision-making**. Learning-based policies can usually achieve good average performance but suffer from unbounded worst-case performance. To meet the requirements for the worst-case performance of learning-based policies, learning-augmented algorithms are developed for online control/optimization problems [38, 51, 17, 35, 34]. To guarantee the performance for each problem instance, learning-augmented algorithm can perform an online switch between ML policy and prior [51], combine the ML policy and prior with an adaptive parameter [38], or project the ML actions into safe action sets relying on the prior actions [35]. Compared with learning-augmented algorithms, we consider more general online settings without knowing the exact dynamic model. Also, our problem can guarantee the cost performance for each round in any episode compared with a policy prior, which has not been studied by existing learning-augmented algorithms.

# 3 Problem Formulation

## 3.1 Anytime-Competitive MDP

In this section, we introduce the setting of a novel MDP problem called Anytime-Competitive Markov Decision Process (A-CMDP), denoted as $\mathcal{M}(\mathcal{X}, \mathcal{A}, \mathcal{F}, g, H, r, c, \pi, \pi^\dagger)$. In A-CMDP, each episode has $H$ rounds. The state at each round is denoted as $x_h \in \mathcal{X}, h \in [H]$. At each round of an episode, the agent selects an action $a_h$ from an action set $\mathcal{A}$. The environment generates a reward $r_h(x_h, a_h)$ and a cost $c_h(x_h, a_h)$ with $r_h \in \mathcal{R}$ and $c_h \in \mathcal{C}$. We model the dynamics as $x_{h+1} = f_h(x_h, a_h)$ where $f_h \in \mathcal{F}$ is a random transition function drawn from an unknown distribution $g(f_h)$ with the density $g \in \mathcal{G}$. The agent has no access to the random function $f_h$ but can observe the state $x_h$ at each round $h$. Note that we model the dynamics in a function style for ease of presentation, and this dynamic model can be translated into the transition probability in standard MDP models [59, 5] as $\mathbb{P}(x_{h+1} \mid x_h, a_h) = \sum_{f_h} \mathbb{1}(f_h(x_h, a_h) = x_{h+1})g(f_h)$. A policy $\pi$ is a function which gives the action $a_h$ for each round $h \in [H]$. Let $V_h^\pi(x) = \mathbb{E}\left[\sum_{i=h}^H r_i(x_i, a_i)) \mid x_h = x\right]$ denote the expected value of the total reward from round $h$ by policy $\pi$. One objective of A-CMDP is to maximize the expected total reward starting from the first round which is denoted as $\mathbb{E}_{x_1}[V_1^\pi(x_1)] = \mathbb{E}\left[\sum_{h=1}^H r_h(x_h, a_h))\right]$.

Besides optimizing the expected total reward as in existing MDPs, A-CMDP also guarantees the anytime competitive cost constraints compared with a policy prior $\pi^\dagger$. The policy prior can be a policy that has verified cost performance in real systems or a heuristic policy with strong empirically-guaranteed cost performance, for which concrete examples will be given in the next section. Denote $y_h = (f_h, c_h, r_h)$, and $y_{1:H} = \{y_h\}_{h=1}^H \in \mathcal{Y} = \mathcal{F} \times \mathcal{R} \times \mathcal{C}$ is a sampled sequence of the models in an A-CMDP. Let $J_h^\pi(y_{1:H}) = \sum_{i=1}^h c_i(x_i, a_i)$ be the cost up to round $h \in [H]$ with states $x_i, i \in [h]$ and actions $a_i, i \in [h]$ of a policy $\pi$. Also, let $J_h^{\pi^\dagger}(y_{1:H}) = \sum_{i=1}^h c_i(x_i^\dagger, a_i^\dagger)$ be the cost of the prior with states $x_i^\dagger, i \in [h]$ and actions $a_i^\dagger, i \in [h]$ of the prior $\pi^\dagger$. The anytime competitive constraints are defined as below.

**Definition 3.1** (Anytime competitive constraints). If a policy $\pi$ satisfies $(\lambda, b)-$anytime competitiveness, the cost of $\pi$ never exceeds the cost of the policy prior $\pi^\dagger$ relaxed by parameters $\lambda \geq 0$ and $b \geq 0$, i.e. for any round $h$ in any model sequence $y_{1:H} \in \mathcal{Y}$, it holds that $J_h^\pi(y_{1:H}) \leq (1+\lambda)J_h^{\pi^\dagger}(y_{1:H})+hb$.

Now, we can formally express the objective of A-CMDP with $\Pi$ being the policy space as

$$\max_{\pi \in \Pi} \mathbb{E}_{x_1}\left[V_1^\pi(x_1)\right], \quad s.t. \; J_h^\pi(y_{1:H}) \leq (1 + \lambda)J_h^{\pi^\dagger}(y_{1:H}) + hb, \quad \forall h \in [H], \forall y_{1:H} \in \mathcal{Y}. \quad (1)$$

Let $\Pi_{\lambda,b}$ be the collection of policies that satisfy the anytime competitive constraints in (1). We design an anytime-competitive RL algorithm that explores the policy space $\Pi_{\lambda,b}$ in $K$ episodes to optimize the expected reward $\mathbb{E}_{x_1}[V_1^\pi(x_1)]$. Note that different from constrained/conservative MDPs [26, 29, 61, 68, 1, 67], the anytime competitive constraints in (1) must be satisfied for any round in any sampled episode $y_{1:H} \in \mathcal{Y}$ given relaxed parameters $\lambda, b \geq 0$. To evaluate the performance of the learned policy $\pi^k \in \Pi_{\lambda,b}, k \in [K]$ and the impact of the anytime competitive constraints, we consider the regret performance metric defined as

$$\text{Regret}(K) = \sum_{k=1}^{K} \mathbb{E}_{x_1}\left[V_1^{\pi^*}(x_1) - V_1^{\pi^k}(x_1)\right], \text{with } \pi^k \in \Pi_{\lambda,b} \quad (2)$$

where $\pi^* = \arg\max_{\pi \in \Pi} \mathbb{E}_{x_1}[V_1^\pi(x_1)]$ is an optimal policy without considering the anytime competitive constraints. When $\lambda$ or $b$ becomes larger, the constraints get less strict and the algorithm has more flexibility to minimize the regret in (2). Thus, the regret analysis will show the trade-off between optimizing the expected reward and satisfying the anytime cost competitiveness.

In this paper, we make additional assumptions on the cost functions, transition functions, and the prior policy which are important for the anytime-competitive algorithm design and analysis.

**Assumption 3.2.** All the cost functions in the space $\mathcal{C}$ have a minimum value $\epsilon \geq 0$, i.e. $\forall(x, a), \forall h \in [H], c_h(x, a) \geq \epsilon \geq 0$, and are $L_c$-Lipschitz continuous with respect to action $a_h$ and the state $x_h$. All the transition functions in the space $\mathcal{F}$ are $L_f$-Lipschitz continuous with respect to action $a_h$ and the state $x_h$. The parameters $\epsilon, L_c$ and $L_f$ are known to the agent.

The Lipschitz continuity of cost and transition functions can also be found in other works on model-based MDP [45, 5, 28]. The Lispchitz assumptions actually apply to many continuous mission-critical systems like cooling systems [16], power systems [54, 15] and carbon-aware datacenters [49]. In these systems, the agents have no access to concrete cost and transition functions, but they can evaluate the Lipschitz constants of cost and dynamic functions based on the prior knowledge of the systems. The minimum cost value can be as low as zero, but the knowledge of a positive minimum cost $\epsilon$ can be utilized to improve the reward performance which will be discussed in Section 5.1.

**Definition 3.3** (Telescoping policy). A policy $\pi$ satisfies the telescoping property if the policy is applied from round $h_1$ to $h_2$ with initialized states $x_{h_1}$ and $x'_{h_1}$, it holds for the corresponding states $x_{h_2}$ and $x'_{h_2}$ at round $h_2$ that

$$\|x_{h_2} - x'_{h_2}\| \leq p(h_2 - h_1)\|x_{h_1} - x'_{h_1}\|, \quad (3)$$

where $p(h)$ is called a perturbation function with $h$ and $p(0) = 1$.

**Assumption 3.4.** The prior policy $\pi^\dagger$ satisfies the telescoping property with some perturbation function $p$. Furthermore, $\pi^\dagger$ is Lipschitz continuous.

The telescoping property in Definition 3.3 indicates that with an initial state perturbation at a fixed round, the maximum divergence of the states afterwards is bounded. Thus, the perturbation function $p$ measures the sensitivity of the state perturbation with respect to a policy prior. The telescoping property is satisfied for many policy priors [60, 43]. It is also assumed for perturbation analysis in model predictive control [42].

Note that in A-CMDP, the constraints are required to be satisfied for any round in any sequence, which is much more stringent than constraint satisfaction in expectation or with a high probability. In fact, the any-time constraints cannot be theoretically guaranteed without further knowledge on the system [8, 56]. This paper firstly shows that Assumption 3.2 and Assumption 3.4, which are reasonable for many mission-critical applications [16, 54, 15, 49], are enough to guarantee the anytime competitive constraints, thus advancing the deployment of RL in mission-critical applications.

## 3.2 Motivating Examples

The anytime competitiveness has direct motivations from many mission-critical control systems. We present two examples in this section and defer other examples to the appendix.

**Safe cooling control in data centers.** In mission-critical infrastructures like data centers, the agent needs to make decisions on cooling equipment management to maintain a temperature range and achieve a high energy efficiency. Over many years, rule-based policies have been used in cooling systems and have verified cooling performance in maintaining a suitable temperature for computing [46]. Recently, RL algorithms are developed for cooling control in data centers to optimize the energy efficiency [63, 16, 46]. The safety concerns of RL policies, however, hinder their deployment in real systems. In data centers, an unreliable cooling policy can overheat devices and denial critical services, causing a huge loss [19, 46]. The safety risk is especially high at the early exploration stage of RL in the real environment. Therefore, it is crucial to guarantee the constraints on cooling performance at anytime in any episode for safety. With the reliable rule-based policies as control priors, A-CMDP can accurately model the critical parts of the cooling control problem, opening a path towards learning reliable cooling policies for data centers.

**Workload scheduling in carbon-intelligent computing.** The world is witnessing a growing demand for computing power due to new computing applications. The large carbon footprint of computing has become a problem that cannot be ignored [49, 64, 52, 23]. Studies find that the amount of carbon emission per kilowatt-hour on electricity grid varies with time and locations due to the various types of electricity generation [33, 32, 9]. Exploiting the time-varying property of carbon efficiency, recent studies are developing workload scheduling policies (e.g. delay some temporally flexible workloads) to optimize the total carbon efficiency [49]. However, an unreliable workload scheduling policy in data centers can cause a large computing latency, resulting in an unsatisfactory Quality of Service (QoS). Thus, to achieve a high carbon efficiency while guaranteeing a low computing latency, we need to solve an A-CMDP which leverages RL to improve the carbon efficiency while guaranteeing the QoS constraints compared with a policy prior targeting at computing latency [20, 30, 12, 72, 71]. This also resembles the practice of carbon-intelligent computing adopted by Google [49].

# 4 Methods

In this section, we first propose an algorithm to guarantee the anytime competitive constraints for any episode, and then give an RL algorithm to achieve a high expected reward under the guarantee of the anytime competitive constraints.

## 4.1 Guarantee the Anytime Constraints

It is challenging to guarantee the anytime competitive constraints in (1) for an RL policy in any episode due to the following. First of all, in MDPs, the agent can only observe the *real* states $\{x_h\}_{h=1}^H$ corresponding to the truly-selected actions $\{a_h\}_{h=1}^H$. The agent does not select the actions $a_h^\dagger$ of the prior, so the states of the prior $x_h^\dagger$ are *virtual* states that are not observed. Thus, the agent cannot evaluate the prior cost $J_h^{\pi^\dagger}$ which is in the anytime competitive constraint at each round $h$. Also, the action at each round $h$ has an impact on the costs in the future rounds $i, i > h$ based on the random transition models $f_i, i \geq h$. Thus, besides satisfying the constraints in the current round, we need to have a good planning for the future rounds to avoid any possible constraint violations even without the exact knowledge of transition and/or cost models. Additionally, the RL policy may be arbitrarily bad in the environment and can give high costs (especially when very limited training data is available), making constraint satisfaction even harder.

Despite the challenges, we design safe action sets $\{\mathcal{A}_h, h \in [H]\}$ to guarantee the anytime competitive constraints: if action $a_h$ is strictly selected from $\mathcal{A}_h$ for each round $h$, the anytime competitive constraints for all rounds are guaranteed. As discussed above, the anytime competitive constraints cannot be evaluated at any time since the policy prior's state and cost information is not available. Thus, we propose to convert the original anytime competitive constraints into constraints that only depend on the known parameters and the action differences between the real policy and the policy prior. We give the design of the safe action sets based on the next proposition. For the ease of presentation, we denote $c_i = c_i(x_i, a_i)$ as the real cost and $c_i^\dagger = c_i(x_i^\dagger, \pi(x_i^\dagger))$ as the cost of the policy prior at round $i$.

**Proposition 4.1.** *Suppose that Assumption 3.2 and 3.4 are satisfied. At round $h$ with costs $\{c_i\}_{i=1}^{h-1}$ observed, the anytime competitive constraints $J_{h'}^{\pi} \leq (1+\lambda)J_{h'}^{\pi^\dagger} + h'b$ for rounds $h' = h, \cdots, H$ are satisfied if for all subsequent rounds $h' = h, \cdots, H$,*

$$\sum_{j=h}^{h'} \Gamma_{j,j}\|a_j - \pi^\dagger(x_j)\| \leq G_{h,h'}, \ \forall h' = h, \cdots, H, \tag{4}$$

*where $\Gamma_{j,n} = \sum_{i=n}^{H} q_{j,i}, (j \in [H], \forall n \geq j)$, with $q_{j,i} = L_c \mathbb{1}(j = i) + L_c(1 + L_{\pi^\dagger})L_f p(i - 1 - j)\mathbb{1}(j < i), (\forall j \in [H], i \geq j)$, relying on known parameters, and $G_{h,h'}$ is called the allowed deviation which is expressed as*

$$G_{h,h'} = \sum_{i=1}^{h-1} \left( (1+\lambda)\hat{c}_i^\dagger - c_i - \Gamma_{i,h}d_i \right) + (h' - h + 1)(\lambda\epsilon + b), \tag{5}$$

*where $\hat{c}_i^\dagger = \max \left\{ \epsilon, c_i - \sum_{j=1}^{i} q_{j,i}d_j \right\}, (\forall i \in [H])$, is the lower bound of of $c_i^\dagger$, and $d_j = \|a_j - \pi^\dagger(x_j)\|, \forall j \in [H]$ is the action difference at round $j$.*

At each round $h \in [H]$, Proposition 4.1 provides a sufficient condition for satisfying all the anytime competitive constraints from round $h$ to round $H$ given in (1). The meanings of the parameters in Proposition 4.1 are explained as follows. The weight $q_{j,i}$ measures the impact of action deviation at round $j$ on the cost difference $|c_i - c_i^\dagger|$ at round $i \geq j$, and the weight $\Gamma_{j,n}$ indicates the total impact of the action deviation at round $j$ on the sum of the cost differences from rounds $n$ to round $H$. Based on the definition of $q_{j,i}$, we get $\hat{c}_i^\dagger$ as a lower bound of the prior cost $c_i^\dagger$. With these bounds, we can calculate the maximum allowed total action deviation compared with the prior actions $\pi^\dagger(x_j)$ from round $j = h$ to $h'$ as $G_{h,h'}$

By applying Proposition 4.1 at initialization, we can guarantee the anytime competitive constraints for all rounds $h' \in [H]$ if we ensure that for all rounds $h' \in [H]$, $\sum_{j=1}^{h'} \Gamma_{j,j}\|a_j - \pi^\dagger(x_j)\| \leq G_{1,h'} = h'(\lambda\epsilon + b)$. This sufficient condition is a long-term constraint relying on the relaxation parameters $\lambda$ and $b$. Although we can guarantee the anytime competitive constraints by the sufficient condition obtained at initialization, we apply Proposition 4.1 at all the subsequent rounds with the cost feedback information to get larger action sets and more flexibility to optimize the average reward. In this way, we can update the allowed deviation according to the next corollary.

**Corollary 4.2.** *At round 1, we initialize the allowed deviation as $D_1 = \lambda\epsilon + b$. At round $h, h > 1$, the allowed deviation is updated as*

$$D_h = \max \left\{ D_{h-1} + \lambda\epsilon + b - \Gamma_{h-1,h-1}d_{h-1}, \ R_{h-1} + \lambda\epsilon + b \right\} \tag{6}$$

*where $R_{h-1} = \sum_{i=1}^{h-1} \left( (1+\lambda)\hat{c}_i^\dagger - c_i - \Gamma_{i,h}d_i \right)$ with notations defined in Proposition 4.1. The $(\lambda, b)$−anytime competitive constraints in Definition 3.1 are satisfied if it holds at each round $h$ that $\Gamma_{h,h}\|a_h - \pi^\dagger(x_h)\| \leq D_h$.*

Corollary 4.2 gives a direct way to calculate the allowed action deviation at each round. In the update rule (6) of the allowed deviation, the first term of the maximum operation is based on the deviation calculation at round $h - 1$ while the second term is obtained by applying Proposition 4.1 for round $h$.

We can find that the conditions to satisfy the anytime competitive constraints can be controlled by parameters $\lambda$ and $b$. With larger $\lambda$ and $b$, the anytime competitive constraints are relaxed and the conditions in Corollary 4.2 get less stringent. Also, the conditions in Corollary 4.2 rely on the minimum cost value $\epsilon$ and other system parameters including Lipschitz constants $L_c, L_f, L_{\pi^\dagger}$ and telescoping parameters $p$ through $\Gamma_{h,h}$. Since $\Gamma_{h,h}$ increases with the Lipschitz and telescoping parameters, even if the estimated Lipschitz constants and the telescoping parameters are higher than the actual values or the estimated minimum cost is lower than the actual value, the obtained condition by Corollary 4.2 is sufficient to guarantee the anytime competitive constraints, although it is more stringent than the condition calculated by the actual parameters.

By Corollary 4.2, we can define the safe action set at each round $h$ as

$$\mathcal{A}_h(D_h) = \left\{ a \mid \Gamma_{h,h}\|a - \pi^\dagger(x_h)\| \leq D_h \right\}. \tag{7}$$

---

**Algorithm 1** Anytime-Competitive Decision-making (`ACD`)

---

**Initialization:** Initialize an allowed deviation: $D_1 = \lambda\epsilon + b$.
**for** $h = 1, \cdots, H$ **do**
    Obtain the output of the ML policy $\tilde{\pi}$ as $\tilde{a}_h$.
    Select the action $a_t$ by projecting $\tilde{a}_h$ into the safe action set $\mathcal{A}_h(D_h)$ in (7).
    Update the allowed deviation $D_{h+1}$ by (6).
**end for**

---

With the safe action set design in (7), we propose a projection-based algorithm called `ACD` in Algorithm 1. We first initialize an allowed deviation as $D_1 = \lambda\epsilon + b$. When the output $\tilde{a}_h$ of the ML model is obtained at each round $h$, it is projected into a safe action set $\mathcal{A}_h(D_h)$ depending on the allowed deviation $D_h$, i.e. $a_h = P_{\mathcal{A}_h(D_h)}(\tilde{a}_h) = \arg\min_{a\in\mathcal{A}_h(D_h)} \|a - \tilde{a}_h\|$. The projection can be efficiently solved by many existing methods on constrained policy learning [67, 11, 4, 41, 24]. The allowed deviation is then updated based on Corollary 4.2. Intuitively, if the actions are closer to the prior actions before $h$, i.e. the action deviations $\{d_i\}_{i=1}^{h-1}$ get smaller, then $R_{h-1}$ becomes larger and $D_h$ becomes larger, leaving more flexibility to deviate from $a_i^\dagger, i \geq h$ in subsequent rounds.

### 4.2 Anytime-Competitive RL

The anytime competitive constraints have been satisfied by Algorithm 1, but it remains to design an RL algorithm to optimize the average reward under the anytime competitive cost constraints, which is given in this section.

The anytime-competitive decision-making algorithm in Algorithm 1 defines a new MDP, with an additional set of allowed deviations $\mathcal{D}$ to the A-MDP defined in Section 3.1, denoted as $\tilde{\mathcal{M}}(\mathcal{X}, \mathcal{D}, \mathcal{A}, \mathcal{F}, g, H, r, c, \tilde{\pi}, \pi^\dagger)$. In the new MDP, we define an augmented state $s_h$ which include the original state $x_h$, the allowed deviation $D_h \in \mathcal{D}$, and history information $\{c_i\}_{i=1}^{h-1}$ and $\{d_i\}_{i=1}^{h-1}$. The transition of $x_h$ is defined by $f_h$ in Section 3.1 and needs to be learned while the transition of $D_h$ is defined in (6) and is known to the agent. The ML policy $\tilde{\pi}$ gives an output $\tilde{a}_h$ and the selected action is the projected action $a_h = P_{\mathcal{A}_h(D_h)}(\tilde{a}_h)$. Then the environment generates a reward $r_h(x_h, P_{\mathcal{A}_h(D_h)}(\tilde{a}_h))$ and a cost $c_h(x_h, P_{\mathcal{A}_h(D_h)}(\tilde{a}_h))$. Thus, the value function corresponding to the ML policy $\tilde{\pi}$ can be expressed as $\tilde{V}_h^{\tilde{\pi}}(s_h) = \mathbb{E}\left[\sum_{i=h}^H r_i(x_i, P_{\mathcal{A}_h(D_h)}(\tilde{a}_h))\right]$ with $\tilde{a}_h$ being the output of the ML policy $\tilde{\pi}$. For notation convenience, we sometimes write the actions of $\pi^*$ and $\pi^\dagger$ as $\pi^*(s)$ and $\pi^\dagger(s)$ even though they only reply on the original state $x$ in $s$.

To solve the MDP, we propose a model-based RL algorithm called `ACRL` in Algorithm 2. Different from the existing model-based RL algorithms [48, 6, 70], `ACRL` utilizes the dynamic model of A-CMDP and `ACD` (Algorithm 1) to optimize the average reward. Given a transition distribution $g$ at episode $k$, we perform value iteration to update $\tilde{Q}$ functions for $h = 1, \cdots, H$.

$$\tilde{Q}_h^k(s_h, \tilde{a}_h) = r_h(x_h, a_h) + \mathbb{E}_g\left[\tilde{V}_{h+1}^k(s_{h+1}) \mid s_h, a_h\right], \quad \tilde{V}_h^k(s_h) = \max_{a\in\mathcal{A}} \tilde{Q}_h^k(s_h, a),$$

$$\mathbb{E}_g\left[\tilde{V}_{h+1}^k(s_{h+1}) \mid s_h, a_h\right] = \sum_{f\in\mathcal{F}} \tilde{V}_{h+1}^k(s_{h+1})g(f), \tag{8}$$

where $a_h = P_{\mathcal{A}_h(D_h)}(\tilde{a}_h)$, $\tilde{Q}_{H+1,k}(s, a) = 0$, $\tilde{V}_{H+1,k}(s) = 0$. The transition model $g$ is estimated as

$$\hat{g}^k = \arg\min_{g\in\mathcal{G}} \sum_{i=1}^{k-1}\sum_{h=1}^H \left(\mathbb{E}_g\left[\tilde{V}_{h+1}^i(s_{h+1}) \mid s_h, a_h\right] - \tilde{V}_{h+1}^i(s_{h+1})\right)^2. \tag{9}$$

Based on the transition estimation, we can calculate the confidence set of the transition model as

$$\mathcal{G}_k = \left\{g\in\mathcal{G} \;\middle|\; \left|\sum_{i=1}^{k-1}\sum_{h=1}^H \left(\mathbb{E}_g\left[\tilde{V}_{h+1}^i(s_{h+1}) \mid s_h, a_h\right] - \mathbb{E}_{\hat{g}^k}\left[\tilde{V}_{h+1}^i(s_{h+1}) \mid s_h, a_h\right]\right)^2 \leq \beta_k\right.\right\}, \tag{10}$$

where $\beta_k > 0$ is a confidence parameter.

---

**Algorithm 2** Anytime-Competitive Reinforcement Learning (ACRL)

---
1: **Initialization:** Transition model set $\mathcal{G}_1 = \{\hat{g}^1\}$.
2: **for** each episode $k = 1, \cdots, K$ **do**
3:     Observe the initial state $s_1^k$.
4:     Select $g^k = \arg\max_{g \in \mathcal{G}^k} \mathbb{E}_g\left[V_1(s_1^k)\right]$.
5:     Perform value iteration with $g^k$ in Eqn. (8) and update $\tilde{Q}$ functions $\tilde{Q}_1^k \cdots, \tilde{Q}_H^k$.
6:     **for** each round $h = 1, \cdots, H$ **do**
7:         Run ACD (Algorithm 1) by ML policy $\tilde{\pi}^k(s_h) = \arg\max_{a \in \mathcal{A}} \tilde{Q}_h^k(s_h, a)$
8:         Observe state $s_{h+1}^k$ and store values $\tilde{V}_{h+1}^k(s_{h+1}^k)$.
9:     **end for**
10:    Update transition model $\hat{g}^{k+1}$ using (9) and calculate confidence set $\mathcal{G}_{k+1}$.
11: **end for**

---

With a learned ML policy $\tilde{\pi}^k$ at each episode $k$, the policy used for action selection is the ACD policy $\pi^k$. Given the optimal ML policy $\tilde{\pi}^* = \arg\max_{\tilde{\pi} \in \tilde{\Pi}} \tilde{V}_1^{\tilde{\pi}}(s_1)$ with $\tilde{\Pi}$ being the ML policy space, the optimal ACD policy is denoted as $\pi^\circ$. For state $s_h$ at round $h$, $\pi^k$ and $\pi^\circ$ select actions as

$$\pi^k(s_h) = P_{\mathcal{A}_h(D_h)}(\tilde{\pi}^k(s_h)), \ \pi^\circ(s_h) = P_{\mathcal{A}_h(D_h)}(\tilde{\pi}^*(s_h)). \tag{11}$$

In the definition of A-CMDP, the dimension of the augmented state $s_h$ increases with the length of the horizon $H$, which cloud cause a scalability issue for implementation. The scalability issues also exit in other RL works with history-dependent states [58, 14]. In practice, tractable methods can be designed through feature aggregation [58] or PODMP [66].

## 5   Performance Analysis

In this section, we analyze the reward regret of ACRL to show the impacts of anytime cost constraints on the average reward.

### 5.1   Regret due to Constraint Guarantee

Intuitively, due to the anytime competitive constraints in Eqn. (1), there always exists an unavoidable reward gap between an ACD policy and the optimal-unconstrained policy $\pi^*$. In this section, to quantify this unavoidable gap, we bound the regret of the optimal ACD policy $\pi^\circ$, highlighting the impact of anytime competitive cost constraints on the average reward performance.

**Theorem 5.1.** *Assume that the optimal-unconstrained policy $\pi^*$ has a value function $Q_h^{\pi^*}(x, a)$ which is $L_{Q,h}$-Lipschitz continuous with respect to the action $a$ for all $x$. The regret between the optimal ACD policy $\pi^\circ$ that satisfies $(\lambda, b)-$anytime competitiveness and the optimal-unconstrained policy $\pi^*$ is bounded as*

$$\mathbb{E}_{x_1}\left[V_1^{\pi^*}(x_1) - V_1^{\pi^\circ}(x_1)\right] \leq \mathbb{E}_{y_{1:H}}\left\{\sum_{h=1}^{H} L_{Q,h}\left[\eta - \frac{1}{\Gamma_{h,h}}(\lambda\epsilon + b + \Delta G_h)\right]^+\right\}, \tag{12}$$

*where $\eta = \sup_{x \in \mathcal{X}} \|\pi^*(x) - \pi^\dagger(x))\|$ is the maximum action discrepancy between the policy prior $\pi^\dagger$ and optimal-unconstrained policy $\pi^*$; $\Gamma_{h,h}$ is defined in Proposition 4.1; $\Delta G_h = [R_{h-1}]^+$ is the gain of the allowed deviation by applying Proposition 4.1 at round $h$.*

The regret bound stated in Theorem 5.1 is intrinsic and inevitable, due to the committed assurance of satisfying the anytime competitive constraints. Such a bound cannot be improved via policy learning, i.e., converge to 0 when the number of episodes $K \rightarrow \infty$. This is because to satisfy the $(\lambda, b)-$anytime competitiveness, the feasible policy set $\Pi_{\lambda,b}$ defined under (1) is a subset of the original policy set $\Pi$, and the derived regret is an upper bound of $\max_{\pi \in \Pi} \mathbb{E}_{x_1}\left[V_1^\pi(x_1)\right] - \max_{\pi \in \Pi_{\lambda,b}} \mathbb{E}_{x_1}\left[V_1^\pi(x_1)\right]$. Moreover, the regret bound relies on the action discrepancy $\eta$. This is because if the optimal-unconstrained policy $\pi^*$ is more different from the prior $\pi^\dagger$, its actions are altered to a larger extent to guarantee the constraints, resulting in a larger degradation of the reward performance. More importantly, the regret bound indicates the trade-off

between the reward optimization and anytime competitive constraint satisfaction governed by the parameters $\lambda$ and $b$. When $\lambda$ or $b$ becomes larger, we can get a smaller regret because the anytime competitive constraints in (1) are relaxed to have more flexibility to optimize the average reward. In the extreme cases when $\lambda$ or $b$ is large enough, all the policies in $\Pi$ can satisfy the anytime competitive constraints, so we can get zero regret.

Moreover, the regret bound shows that the update of allowed deviation by applying Proposition 4.1 based on the cost feedback at each round will benefit the reward optimization. By the definition of $R_{h-1}$ in Corollary 4.2, if the real actions deviate more from the prior actions before $h$, the gain $\Delta G_i$ for $i \geq h$ can be smaller, so the actions must be closer to the prior actions in the subsequent rounds, potentially causing a larger regret. Thus, it is important to have a good planing of the action differences $\{d_i\}_{i=1}^{H}$ to get larger allowed action deviations for reward optimization. Exploiting the representation power of machine learning, ACRL can learn a good planning of the action differences, and the ACD policy $\pi^\circ$ corresponding to the optimal ML policy $\tilde{\pi}^*$ can achieve the optimal planing of the action differences.

Last but not least, Theorem 5.1 shows the effects of the systems parameters in Assumption 3.2 and Assumption 3.4 on the regret through $\Gamma_{h,h}$ defined in Proposition 4.1 and the minimum cost $\epsilon$. Observing that $\Gamma_{h,h}$ increases with the systems parameters including the Lipschitz parameters $L_f, L_c, L_{\pi^\dagger}$ and telescoping parameters $p$, a higher estimation of the Lipschitz parameters and telescoping parameters can cause a higher regret. Also, a lower estimation of the minimum cost value can cause a higher regret. Therefore, although knowing the upper bound of the Lipschitz parameters and telescoping parameters and the lower bound of the minimum cost value is enough to guarantee the anytime competitive cost constraints by Proposition 4.1, a lower reward regret can be obtained with a more accurate estimation of these system parameters.

## 5.2 Regret of ACRL

To quantify the regret defined in Eqn. (2), it remains to bound the reward gap between the ACD policy $\pi^k$ and the optimal ACD policy $\pi^\circ$. In this section, we show that $\pi^k$ by ACRL approaches the optimal one $\pi^\circ$ as episode $K \to \infty$ by bounding the pseudo regret

$$\text{PReg}(K) = \mathbb{E}_{x_1} \left[ \sum_{k=1}^{K} \left( V_1^{\pi^\circ}(s_1) - V_1^{\pi^k}(s_1) \right) \right]. \tag{13}$$

**Theorem 5.2.** *Assume that the value function is bounded by $\bar{V}$. Denote a set of function as*

$$\mathcal{Q} = \{q \mid \exists g \in \mathcal{G}, \forall (s,a,v) \in \mathcal{S} \times \mathcal{A} \times \mathcal{V}, q(s,a,v) = \mathbb{E}_{f \sim g}\left[v(s') \mid s,a\right]\}. \tag{14}$$

*If $\beta_k = 2(\bar{V}H)^2 \log\left(\frac{2\mathcal{N}(\mathcal{Q},\alpha,\|\cdot\|_\infty)}{\delta}\right) + C\bar{V}H$ with $\alpha = 1/(KH\log(KH/\delta))$, $C$ being a constant, and $\mathcal{N}(\mathcal{Q},\alpha,\|\cdot\|_\infty)$ being the covering number of $\mathcal{Q}$, with probability at least $1-\delta$, the pseudo regret of Algorithm 2 is bounded as*

$$\text{PReg}(K) \leq 1 + d_{\mathcal{Q}}H\bar{V} + 4\sqrt{d_{\mathcal{Q}}\beta_K KH} + H\sqrt{2KH\log(1/\delta)}, \tag{15}$$

*where $d_{\mathcal{Q}} = \dim_E(\mathcal{Q}, \frac{1}{KH})$ is the Eluder dimension of $\mathcal{Q}$ defined in [50].*

Theorem 5.2 bounds the pseudo regret for each episode $k$. The confidence parameter $\beta_k$ to balance the exploration and exploitation is chosen to get the pseudo regret bound as shown in Theorem 5.2. A higher $\beta_k$ is chosen to encourage the exploration if the covering number of the function space $\mathcal{Q}$, the episode length, or the maximum value becomes larger. Also, the pseudo regret relies on the size of the function space $\mathcal{Q}$ through $d_{\mathcal{Q}}$ and $\beta_K$. With smaller $\lambda$ or $b$, less actions satisfy Corollary 4.2 given a state, and so a smaller state-action space $\mathcal{S} \times \mathcal{A}$ is obtained, which results in a smaller size of the function space $\mathcal{Q}$ and thus a smaller regret.

To get more insights, we also present the overall regret bound when the transition model $g$ can be represented by a linear kernel as in [6, 70], i.e. $g(f) = \langle \phi(f), \theta \rangle$ with dimension of $\theta$ as $d_\theta$, the reward regret in Eqn.2 is bounded as

$$\text{Regret}(K) \leq K\mathbb{E}_{y_{1:H}} \left\{ \sum_{h=1}^{H} L_{Q,h} \left[ \eta - \frac{1}{\Gamma_{h,h}}(\lambda\epsilon + b + \Delta G_h) \right]^+ \right\} + \tilde{O}(\sqrt{H^3\bar{V}^2 K\log(1/\delta)}), \tag{16}$$

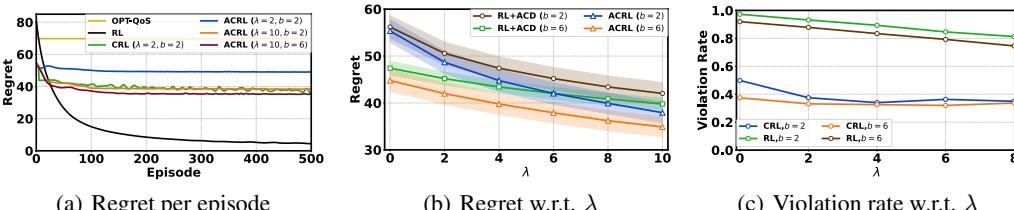

|(a) Regret per episode|(b) Regret w.r.t. $\lambda$|(c) Violation rate w.r.t. $\lambda$|

Figure 1: Regret and cost violation rate of different algorithms. Shadows in Figure 2(b) show the range of the regret.

where $L_{Q,h}$, $\eta$, $\Gamma_{h,h}$, and $\Delta G_h$ are all defined in Theorem 5.1. The overall regret bound is obtained because under the assumption of linear transition kernel, we have $\beta_K = O((\bar{V}H)^2 \log(\frac{1}{\delta}\mathcal{N}(\mathcal{Q}, \alpha, \|\cdot\|_\infty))) = \tilde{O}((\bar{V}H)^2(d_\theta + \log(1/\delta)))$ [6], and the Eluder dimension is $d_\mathcal{Q} = \tilde{O}(d_\theta)$ [50]. Thus the pseudo regret is $\mathrm{PReg}(K) = \tilde{O}(\sqrt{H^3 \bar{V}^2 K \log(1/\delta)})$ which is sublinear in terms of $K$. With the sublinear pseudo regret $\mathrm{PReg}(K)$, the ACD policy $\pi^k$ performs as asymptotically well as the optimal ACD policy $\pi^\circ$ when $K \to \infty$. Combining with the regret of the optimal ACD policy in Theorem 5.1, we can bound the overall regret of ACRL. Since in the definition of regret, ACD policy is compared with the optimal-unconstrained policy $\pi^*$, the regret bound also includes an unavoidable linear term due to the commitment to satisfy the anytime competitive constraints. The linear term indicates the trade-off between the reward optimization and the anytime competitive constraint satisfaction.

## 6 Empirical Results

We experiment with the application of resource management for carbon-aware computing [49] to empirically show the benefits of ACRL. The aim of the problem is to jointly optimize carbon efficiency and revenue while guaranteeing the constraints on the quality-of-service (QoS). In this problem, there exists a policy prior $\pi^\dagger$ which directly optimizes QoS based on estimated models. In our experiment, we apply ACRL to optimize the expected reward and guarantee that the real QoS cost is no worse than that of the policy prior. The concrete settings can be found in Appendix A.

Figure 1(a) gives the regret changing the in first 500 episodes. Figure 1(b) shows the regret with different $\lambda$ and $b$, demonstrating the trade-off between reward optimization and the satisfaction of anytime competitive constraints. Figure 1(c) shows the probability of the violation of the anytime competitive constraints by RL and constrained RL. ACRL and ML models with ACD have no violation of anytime competitive constraints. More analysis about the results are provided in Appendix A due to space limitations.

## 7 Concluding Remarks

This paper considers a novel MDP setting called A-CMDP where the goal is to optimize the average reward while guaranteeing the anytime competitive constraints which require the cost of a learned policy never exceed that of a policy prior $\pi^\dagger$ for any round $h$ in any episode. To guarantee the anytime competitive constraints, we design ACD, which projects the output of an ML policy into a safe action set at each round. Then, we formulate the decision process of ACD as a new MDP and propose a model-based RL algorithm ACRL to optimize the average reward under the anytime competitive constraints. Our performance analysis shows the tradeoff between the reward optimization and the satisfaction of the anytime competitive constraints.

**Future directions.** Our results are based on the assumptions on the Lipschitz continuity of the cost, dynamic functions, and the policy prior, as well as the telescoping properties of the policy prior, which are also supposed or verified in other literature [5, 28, 60, 40]. In addition, to guarantee the anytime competitive constraints, the agent is assumed to have access to the Lipschitz constants, the minimum cost value, and the perturbation function. However, since the anytime competitive constraints are much stricter than the expected constraints or the constraints with a high probability, there is no way to guarantee them without any knowledge of the key properties of a mission-critical system. Our work presents the first policy design to solve A-CMDP, but it would be interesting to design anytime-competitive policies with milder assumptions in the future.

## Acknowledgement

We would like to thank the anonymous reviewers for their helpful comments. Jianyi Yang, Pengfei Li and Shaolei Ren were supported in part by the U.S. NSF under the grant CNS–1910208. Tongxin Li was partially supported by the NSFC grant No. 72301234, the Guangdong Key Lab of Mathematical Foundations for Artificial Intelligence, and the start-up funding UDF01002773 of CUHK-Shenzhen. Adam Wierman was supported in part by the U.S. NSF under grants CNS–2146814, CPS–2136197, CNS–2106403, NGSDI–2105648.

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
