# Anytime-Competitive Reinforcement Learning with Policy Prior – Supplementary Material

We provide the empirical results and proofs of our theorems in the appendix.

## A    Empirical Results - Carbon-Aware Resource Management

### A.1    Problem Formulation

We consider the sustainable workload scheduling problem in datacenters to jointly optimize the carbon efficiency and the revenue while guaranteeing the quality-of-service (QoS). In this problem, the average carbon efficiency and revenue can be optimized while QoS must always be ensured at each step. The agent needs to decide the computing resource $a_h$ measured by energy $(kWh)$ for each round $h$. The state $x_h$ is the remaining demand for each round $h$ and is updated as

$$x_h = f(x_{h-1}, \mu_h, a_h) = [V_x(x_{h-1}) + \mu_h - V_a(a_h)]^+, \tag{17}$$

where $V_x$ is a random function of $x_{h-1}$ measuring the randomly decayed remaining demands (e.g., due to workload dropping), $\mu_h$ is the arrival demand at round $h$, and $V_a$ is a random function in terms of $a_h$ and outputs the amount of processed workload. With the random functions $V_x$ and $V_a$, the remaining workload $x_h$ at round $h$ is drawn from $\mathbb{P}(x_h \mid x_{h-1}, \mu_h, a_h)$. Here, we focus on flexibly deferrable workloads (e.g., model training and batch data processing) [49].

The energy efficiency reward is modeled by a penalty for the carbon footprint at each round. Let $C_h$ be the amount of renewable at round $h$, the energy efficiency reward is expressed as $\text{efficiency}_h = -([a_h - C_h]^+)^2$. The revenue function is modeled as a general power-law function [53] as $\text{revenue}_h = C_r V_a^\alpha(a_h)$ with $\alpha \in (0,1)$. In datacenters, we also need to consider a switching cost $\gamma_2 \|a_h - a_{h-1}\|^2$ at each round $h$ to avoid switching on/off servers frequently. Thus, the reward in this problem is formulated as

$$\text{reward}_h = \text{efficiency}_h + \gamma_1 \cdot \text{revenue}_h - \gamma_2 \cdot \|a_h - a_{h-1}\|^2. \tag{18}$$

Besides the reward, QoS is also crucial for deferrable workloads in datacenters. In this work, we model QoS as a cost function of the remaining demand as follows:

$$cost_{\text{QoS},h} = x_h^\top Q_1 x_h + Q_2^\top x_h + Q_3, \tag{19}$$

where $Q_1$, $Q_2$ and $Q_3$ are constants.

As shown in Eqn. (20), given a baseline $\pi^\dagger$ that has been verified to achieve a satisfactory QoS, our goal is to optimize the expected reward and guarantee the QoS for any time in any sequence, i.e.

$$\max_{\pi \in \Pi} \quad \mathbb{E}\left[\sum_{h=1}^{H} \text{reward}_h\right],$$

$$s.t. \quad \sum_{h=1}^{h'} cost_{\text{QoS},h}(\pi) \leq (1+\lambda)\sum_{h=1}^{h'} cost_{\text{QoS},h}(\pi^\dagger) + h'b, \quad \forall h' \in [H], \tag{20}$$

which is consistent with the definition of anytime competitive constraints in Definition 3.1.

### A.2    Baselines

In the experiments, we consider different baselines as below.

• QoS Optimization (`OPT-QoS`): This baseline policy prior directly optimizes QoS in (19) based on estimated models $\hat{V}_x$ and $\hat{V}_a$. Without taking efficiency or revenue into consideration, `OPT-QoS` essentially always schedules as many computing resources as possible to lower the QoS cost based on the estimated arrival demand.

• Reinforcement Learning (`RL`): This is a model-based reinforcement learning algorithm to optimize the expected reward $\mathbb{E}\left[\sum_{h=1}^{H} \text{reward}_h\right]$ without considering any QoS constraints.

● Constrained Reinforcement Learning (CRL): This is a constrained reinforcement learning to optimize the reward with the expected QoS cost constraint as shown below:

$$\max_{\pi \in \Pi} \quad \mathbb{E}\left[\sum_{h=1}^{H} \text{reward}_h\right], \quad s.t. \quad \mathbb{E}\left[\sum_{h=1}^{H} cost_{\text{QoS},h}(\pi) - (1+\lambda)\sum_{h=1}^{H} cost_{\text{QoS},h}(\pi^\dagger)\right] \leq B. \quad (21)$$

● Random RL policy with ACD (Random +ACD): This algorithm selects actions by ACD in Algorithm 1 with a random ML model $\tilde{\pi}$ as the input of ACD.

● Trained RL policy with ACD (RL +ACD): This algorithm selects actions by ACD in Algorithm 1 with the RL policy trained to optimize the expected reward without accounting for QoS.

● Anytime-Competitive Reinforcement Learning (ACRL): This is the proposed Algorithm 2 which optimizes the expected reward while guaranteeing the anytime competitive QoS cost constraints in (20). In each inference, ACD in Algorithm 1 is used to select actions.

### A.3 Experiment Settings

In the experiments, we evaluate the performances with the following experiment settings.

**System parameters.** We evaluate the regret and the cost constraints for different choices of parameters. The results are given for different anytime competitive constraint parameters including $\lambda$ chosen from $[0, 10]$ and $b$ chosen from $\{2, 6\}$. With smaller $\lambda$ and $b$, we have more stringent constraints, and vice versa. In the experiments, we choose $\alpha = 0.5$ for the revenue function to simulate a typical effect of the scheduled resource on the revenue. To scale different rewards into the same magnitude, we choose the weight for the revenue as $\gamma_1 = 4$, and the weight for the switching cost as $\gamma_2 = 1$. For the QoS cost function, we choose $Q_1 = Q_2 = Q_3 = 1$, so we have the minimum QoS cost as $\epsilon = Q_3 = 1$. The transition model $f$ are from a function space defined by random functions $V_x$ and $V_a$. To create the environment for RL, $V_x(x_h)$ is drawn from a uniform distribution with range $[0.9 \cdot x_h, x_h]$, and $V_a(a_h)$ is drawn from a normal distribution with $0.8 \cdot a_h$ as the center.

**Data.** For experiments, we create an environment based on a renewable dataset and a demand dataset. The renewable dataset is a public dataset from California Independent System Operator [47] which contains the hourly renewable generation in 2019. The renewable sequences from multiple sources (solar, wind, water ) are summed together and scaled to be the values of $\{C_h\}_{h=1}^{H}$ in the problem formulation. In addition, we use the Azure Cloud Dataset [18] as the demand dataset which includes hourly CPU utilization in the same year of 2019. We choose the sequences of the first three months and augment them to 4000 episodes for policy exploration, and we hold out the sequences of the last two months for testing.

**Learning settings.** To ensure fair comparisons, we choose the same neural network architecture as the policy network for different methods. The policy neural network has two hidden layers and each hidden layer has 40 neurons. For training, the policy network parameters are initialized by Gaussian distribution. The reinforcement learning has total $K = 4000$ episodes. We update the neural network every 50 episodes with a weight update rate of $10^{-3}$. We apply Adam optimizer to update the weights of neural networks.

### A.4 Results

We show the empirical results for both regret and QoS cost and discuss the insights from these results.

**Regret evaluation.** The reward regrets as defined in Eqn. (2) are given in Figure 2. To evaluate the regret, we use the RL policy after the exploration for total $4000$ episodes as the optimal RL policy $\pi^*$. The results are given for different anytime competitiveness parameters $\lambda$ and $b$.

Figure 2(a) shows the varying regret of different algorithms for the first 500 episodes. Without including reward as an objective, the policy prior OPT-QoS is an algorithm that is not updated over time and always gives the highest regret. Without the QoS cost constraints, RL approaches the optimal RL policy that can give the best regret as time goes on. The constrained RL (CRL) and anytime-competitive RL (ACRL) are guaranteed to satisfy the expected constraint in (21) and the anytime competitive constraints in (20), respectively, so their reward regrets are higher than RL. Also, we can find that with larger $\lambda$ and/or $b$, ACRL can achieve lower regret after about 200 episodes. This

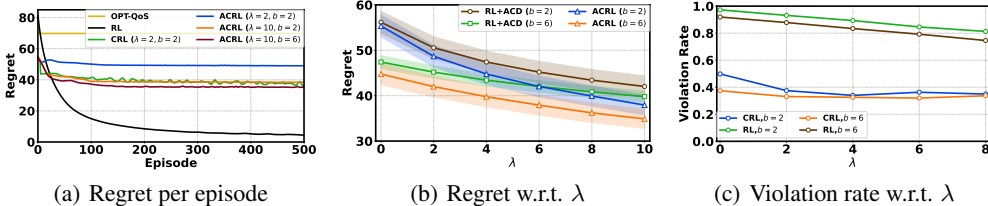

(a) Regret per episode      (b) Regret w.r.t. $\lambda$      (c) Violation rate w.r.t. $\lambda$

Figure 2: Regret and cost violation rate of different algorithms. Figure 2(a) gives the regret changing with episodes. Figure 2(b) shows the regret with different $\lambda$ and $b$ after exploration for all the $4000$ episodes. Shadows in Figure 2(b) show the range of regret. Figure 2(c) shows the probability of the violation of the anytime competitive constraints. Figure 2(c) shows the probability of the violation of the anytime competitive constraints.

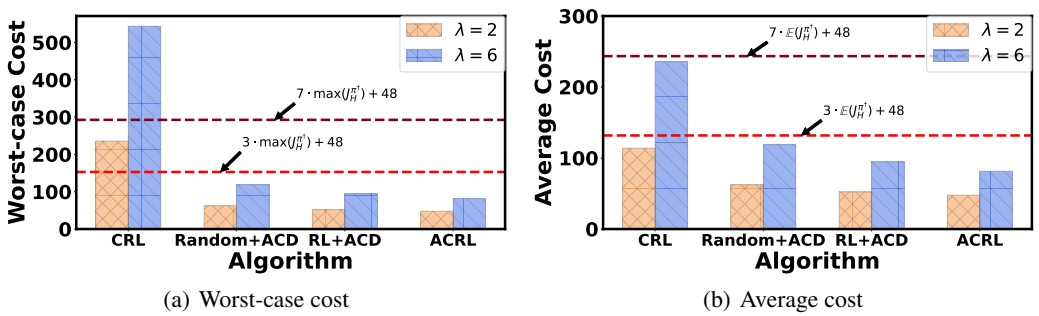

(a) Worst-case cost          (b) Average cost

Figure 3: QoS costs of different algorithms. Figure 3(a) and Figure 3(b) give the worst-case costs and the average cost for different algorithms under $b = 2$, $\lambda = 2$ and $\lambda = 6$, respectively. For `OPT-QoS` $\pi^\dagger$, the worst-case cost $\max(J_H^{\pi^\dagger})$ and average cost $\mathbb{E}(J_H^{\pi^\dagger})$ are 34.95 and 27.92, respectively.

is because the larger $\lambda$ or $b$ gives less stringent anytime competitive constraints and leaves more flexibility to reduce the regret as shown in Theorem 5.1. Moreover, we can observe that `ACRL` with smaller $\lambda$ and $b$ (e.g. $\lambda = 2, b = 2$ in the figure) converges faster to the optimal policy under the anytime competitive constraints. The reason is that the constraints with smaller $\lambda$ and $b$ provide a smaller policy space to explore, resulting in a smaller Eluder dimension $d_Q$ shown in Theorem 5.2.

In Figure 2(b), we give the optimal regret of the algorithms after enough exploration. `ACRL` and `RL` `+ACD` are different in terms of the RL policy $\tilde{\pi}$ used in Algorithm 1. `Random +ACD` uses randomly initialized RL policy as $\tilde{\pi}$ and has a regret as large as from $61$ to $64.51$, exceeding the limit of the regret axis. The optimal regrets of different algorithms decrease with the parameters $\lambda$ and $b$, which is consistent with the findings in Theorem 5.1. Importantly, we can find that under the same $\lambda$ and $b$, `ACRL` can always improve the regret of `RL +ACD`. This is because `RL` explores the original environment defined in Section 3.1 while `ACRL` explores the environment of the new MDP defined at the beginning of Section 4.2, which is the true environment created by `ACD`. This highlights the advantage of `ACRL` in terms of reducing the reward regret by learning with the awareness of the anytime constraints in the true environment.

**Cost evaluation.** In Figure 2(c), we show the violation probability of the anytime competitive constraints for RL and CRL. Thanks to the theoretical guarantee of the anytime competitive constraints, the algorithms based on `ACD` (`ACRL`, `RL+ACD`) all have zero violation probability, so they are not shown in the figure. Since RL only optimizes the expected reward, the probability of cost constraint violation becomes higher when the competitive constraint becomes more stringent (smaller $\lambda$ or $b$). Although CRL considers the expected cost constraint and achieves a lower violation rate, the violation rate is still not zero since there is no theoretical guarantee of the anytime competitiveness. The violation rate of anytime competitive constraints is not allowed for mission-critical applications, showing the need of an RL algorithm with anytime competitiveness guarantee like `ACRL`.

We evaluate the QoS costs in Figure 3 to verify that the constraints are satisfied.

Figure 3(a) gives the worst-case cost of different algorithms for all the sequences in the testing dataset. The `RL` algorithm without considering the cost objective achieves the worst-case cost of $5015.62$, exceeding the range of the cost axis to a large extent. The dotted horizon lines show the maximum cost bound required by the anytime competitive constraint in Definition 3.1. Clearly, we can find that given any RL policy (even a randomly initialized RL policy) as an input, `ACD` can guarantee the anytime competitive constraints even for the worst-case, but `CRL` fails to guarantee the anytime competitive constraints. In the experiments, we verify that all `ACD` algorithms have zero violation of anytime competitive constraints no matter what ML policy is used, but `CRL` has a violation rate of $27.5\%$ for $\lambda = 2$ and a violation rate of $58.6\%$ for $\lambda = 6$.

Figure 3(b) shows the average cost of different algorithms for all the sequences in the testing dataset. The `RL` algorithm achieves an average QoS cost of $2070.14$, exceeding the range of the cost axis to a large extent. The dotted lines give the maximum average QoS cost bound required by the expected constraint in (21). Since `CRL` is designed to optimize the regret subject to the average QoS constraints in (21), it has no violation in terms of the average QoS cost. The algorithms that use `ACD` guarantees the stricter anytime competitive constraints than the average constraint, so they can also guarantee the average QoS constraint.

## B   Empirical Results - Sustainable AI Inference

### B.1   Problem Formulation

AI tasks are widely deployed on edge datacenters. The renewables are utilized in edge datacenters to reduce the carbon emissions from AI inference. The renewable sources are known for their time-varying and unstable nature. Multiple AI models are often available for a given AI inference service [7]. This provides a flexible balance between accuracy and energy consumption. Thus, the agent needs to decide the model size at each round to optimize the inference performance with a constrained amount of carbon emission. [49, 52].

Specifically, the edge datacenter utilizes a battery to store the renewables and unused energy. The state $x_h$ is the battery state of charge (SoC) at each round $h$. Given an action $a_h$ (the energy consumed by selected models) and renewable $e_h$ at round $h$, the battery SoC is updated as

$$x_h = f(x_{h-1}, e_h, a_h) = [x_{h-1} + V_e(e_h) - V_a(a_h)]^+, \tag{22}$$

where $V_e$ and $V_a$ are random functions with the randomness coming from the charging and discharging rates of different batteries under different temperatures [55]. Since recycled batteries may be used in edge datacenters for sustainability, the charging and discharging functions $V_e(\cdot)$ and $V_a(\cdot)$ are generally unknown given a new problem instance. If at some round $h$, the consumption is larger than the sum of the renewable replenishment and remaining battery energy, i.e. $x_{h-1} + V_e(e_h) < V_a(a_h)$, the fossil energy is used and a cost that penalizes the carbon emission is formulated as

$$cost_{\text{carbon},h} = Q_1 \left\| [V_a(a_h) - x_{h-1} - V_e(e_h)]^+ \right\|^2, \tag{23}$$

where $Q_1$ is a constant.

The reward includes the demand satisfaction revenue and inference performance. With more energy, more demand is satisfied, so we directly penalize the demand that is not served at each round as $\text{reward}_{d,h} = -C_d \cdot ([\mu_h - a_h]^+)^2$ given an AI inference workload $\mu_h$. The inference performance is dependent on the action and is modeled by the log utility to capture the diminishing return, i.e. $\text{reward}_{i,h} = \log(1 + C_i * a_h)$. The total reward for round $h$ is represented as

$$\text{reward}_h = \text{reward}_{d,h} + \gamma_2 \cdot \text{reward}_{i,h} - \gamma_1 \cdot \|a_h - a_{h-1}\|^2 - \gamma_3 \cdot cost_{\text{carbon},h}. \tag{24}$$

Same as the previous problems, given a policy prior $\pi^\dagger$ that balances the minimization of carbon emission and the demand satisfaction, the goal is

$$\max_{\pi \in \Pi} \quad \mathbb{E}\left[ \sum_{h=1}^{H} \text{reward}_h \right],$$

$$s.t. \quad \sum_{h=1}^{h'} cost_{\text{carbon},h}(\pi) \leq (1+\lambda) \sum_{h=1}^{h'} cost_{\text{carbon},h}(\pi^\dagger) + h'b, \quad \forall h' \in [H], \tag{25}$$

## B.2 Settings and Baselines

In the experiments, we consider different baselines as below.

• Carbon Bound (`Carbon-B`): This baseline policy prior makes decisions as below. When the estimated SoC $x_{h-1} + \hat{V}_e(x_{h-1}, e_h) + Q_c$ based on an estimated charging function $\hat{V}_e$ and a slackness $Q_c$ is equal to or higher than the demand $\mu_h$. The action is set as $\mu_h$ to meet the demand. Otherwise, the action is selected to bound the estimated carbon by a positive value $Q_c$ based on an estimated discharging function, i.e. solving $Q_1 \left\| [\hat{V}_a(a_h) - x_{h-1} - \hat{V}_e(e_h)]^+ \right\|^2 = Q_c$.

• Reinforcement Learning (`RL`): This is a model-based reinforcement learning algorithm to optimize the expected reward $\mathbb{E}\left[ \sum_{h=1}^{H} \text{reward}_h \right]$ without considering any cost constraints.

• Constrained Reinforcement Learning (`CRL`): This is a constrained reinforcement learning to optimize the reward with the expected carbon cost constraint as shown below:

$$\max_{\pi \in \Pi} \quad \mathbb{E}\left[ \sum_{h=1}^{H} \text{reward}_h \right], \quad s.t. \quad \mathbb{E}\left[ \sum_{h=1}^{H} cost_{\text{carbon},h}(\pi) - (1+\lambda) \sum_{h=1}^{H} cost_{\text{carbon},h}(\pi^\dagger) \right] \le B. \tag{26}$$

• Anytime-Competitive Reinforcement Learning (`ACRL`): This is the proposed Algorithm 2 which optimizes the expected reward while guaranteeing the anytime competitive carbon cost constraints in (25). In each inference, `ACD` in Algorithm 1 is used to select actions.

We evaluate the methods in the following experiment environments.

**System parameters.** We evaluate the performance for different choices of parameters. The results are given for different anytime competitive constraint parameters including $\lambda$ chosen from $\{5, 7\}$ and $b$ chosen as 6 which controls how stringent the cost constraints are. In the experiments, we convert demand satisfaction reward, the inference performance, the carbon costs and the switching costs into monetary values through parameters $\gamma_1 = 0.5$, $\gamma_2 = 2$, and $\gamma_3 = 0.1$. We choose $Q_1 = 1$ in the carbon cost function. To create the environment for RL, $V_a(a_h)$ is drawn from a uniform distribution with range $[0.7 \cdot a_h, a_h]$, and $V_e(e_h)$ is drawn from a uniform distribution with range $[0.8 \cdot e_h, e_h]$.

**Data.** The inference demand $\mu_t$ comes from the GPU power usage of a large language model [44]. We still use California Independent System Operator [47] dataset to simulate the renewable replenishment for edge data center. We choose the sequences of the first three months and augment them to 2160 episodes for policy training, and we hold out 1440 sequences for testing.

**Learning settings.** The neural network architecture is designed as below. The policy neural network has two hidden layers and each hidden layer has 50 neurons. For training, the policy network parameters are initialized by Gaussian distribution. The reinforcement learning has total $K = 2160$ episodes. We update the neural network every 50 episodes with a weight update rate of $10^{-4}$. We apply Adam optimizer to update the weights of neural networks.

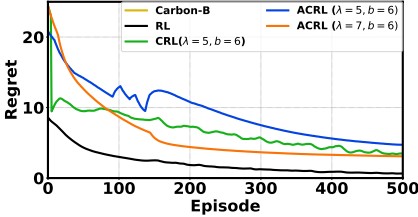

## B.3 Regret Evaluation

Figure 4: Regrets of different algorithms. The regret of `Carbon-B` is 51.417 and is out of the range of y axis.

We give the dynamic reward regret for different algorithms within first 500 episodes in Figure 4. We can find that the regrets of all algorithms decrease as time goes. RL, which directly optimizes the optimal reward without accounting the cost constraint satisfaction, can achieve the lowest reward regret. Both CRL and `ACRL` consider the competitive cost constraints and achieve a larger regret than RL as is indicated by Theorem 5.2. Among them, CRL guarantees the competitive cost constraints in expectation, so it achieves a lower regret than `ACRL` which guarantees the anytime competitive cost constraint with the same parameters $\lambda = 5$ and $b = 6$. However, `ACRL` can theoretically guarantee the anytime competitive constraint, which makes it more suitable for the sustainable AI inference where strict requirements for carbon emission exist.

Moreover, when $\lambda$ increases to 7, we can find that the regret of ACRL is reduced since the anytime competitive constraint is less stringent and there is more flexibility to optimize the expected reward, which demonstrates the trade-off between reward optimization and competitiveness satisfaction given by Theorem 5.1.

# C   Proof of Theorems in Section 4

## C.1   Proof of Proposition 4.1

**Proposition 4.1** *Suppose that Assumption 3.2 and 3.4 are satisfied. At round $h$ with costs $\{c_i\}_{i=1}^{h-1}$ observed, the anytime competitive constraints $J_{h'}^{\pi} \leq (1+\lambda)J_{h'}^{\pi^\dagger} + h'b$ for rounds $h' = h, \cdots, H$ are satisfied if for all subsequent rounds $h' = h, \cdots, H$,*

$$\sum_{j=h}^{h'} \Gamma_{j,j} \|a_j - \pi^\dagger(x_j)\| \leq G_{h,h'}, \forall h' = h, \cdots, H,$$

*where $\Gamma_{j,n} = \sum_{i=n}^{H} q_{j,i}, (j \in [H], \forall n \geq j)$, with $q_{j,i} = L_c \mathbb{1}(j=i) + L_c(1+L_{\pi^\dagger})L_f p(i - 1 - j)\mathbb{1}(j < i), (\forall j \in [H], i \geq j)$, relying on known parameters, and $G_{h,h'}$ is called the allowed deviation which is expressed as*

$$G_{h,h'} = \sum_{i=1}^{h-1} \left( (1+\lambda)\hat{c}_i^\dagger - c_i - \Gamma_{i,h}d_i \right) + (h'-h+1)(\lambda\epsilon + b),$$

*where $\hat{c}_i^\dagger = \max\left\{ \epsilon, c_i - \sum_{j=1}^{i} q_{j,i}d_j \right\}, (\forall i \in [H])$, is the lower bound of of $c_i^\dagger$, and $d_j = \|a_j - \pi^\dagger(x_j)\|, \forall j \in [H]$ is the action difference at round $j$. $\qquad\square$*

*Proof.* First, we bound the state perturbation. As is shown in Figure 5, denote $x_h^{\dagger(i)}$ is the state by applying the policy prior $\pi^{\dagger(i)}$ from round $i$, so the state difference at round $h$ is expressed as

$$
\begin{aligned}
\|x_h - x_h^\dagger\| &= \|\sum_{i=1}^{h-1}(x_h^{\dagger(i+1)} - x_h^{\dagger(i)})\| \\
&\leq \sum_{i=1}^{h-1} \|(x_h^{\dagger(i+1)} - x_h^{\dagger(i)})\| \\
&\leq \sum_{i=1}^{h-1} p(h-1-i)\|(x_{i+1} - x_{i+1}^{\dagger(i)})\| \\
&\leq L_f \sum_{i=1}^{h-1} p(h-1-i)\|a_i - \pi^\dagger(x_i)\|,
\end{aligned}
\tag{27}
$$

where the first inequality holds by triangle inequality, and the second inequality holds by the telescoping property of $\pi^\dagger$.

Then, we bound the gap between the expert cost and true cost at round $h$ as

$$
\begin{aligned}
&|c_h(x_h, a_h) - c_h(x_h^\dagger, a_h^\dagger)| \\
=&c_h(x_h, a_h) - c_h(x_h, \pi^\dagger(x_h)) + c_h(x_h, \pi^\dagger(x_h)) - c_h(x_h^\dagger, \pi^\dagger(x_h^\dagger)) \\
\leq&L_c\|a_h - \pi^\dagger(x_h)\| + L_c(1+L_{\pi^\dagger})\|x_h - x_h^\dagger\| \\
\leq&L_c\|a_h - \pi^\dagger(x_h)\| + L_c(1+L_{\pi^\dagger})L_f \sum_{j=1}^{h-1} p(h-1-j)\|a_j - \pi^\dagger(x_j)\| \\
=&\sum_{j=1}^{h} q_{j,h}\|a_j - \pi^\dagger(x_j)\|,
\end{aligned}
\tag{28}
$$

Figure 5: Illustration of state perturbation

where the first inequality holds by the Lipschitz continuity of cost functions $c_h$ and policy $\pi^\dagger$, the second inequality holds by (27), and $q_{j,h} = L_c \mathbb{1}(j = h) + L_c(1 + L_{\pi^\dagger})L_f p(h - 1 - j)\mathbb{1}(j < h)$.

Recall that the anytime competitive constraint for any round $h' \in [H]$ is $\sum_{i=1}^{h'} c_i(x_i, \pi(x_i)) \leq (1 + \lambda)\sum_{i=1}^{h'} c_i(x_i^\dagger, \pi^\dagger(x_i^\dagger)) + h'b$ which is equivalent to $\sum_{i=1}^{h'} \left( c_i(x_i, \pi(x_i)) - c_i(x_i^\dagger, \pi^\dagger(x_i^\dagger)) \right) \leq \lambda \sum_{i=1}^{h'} c_i(x_i^\dagger, \pi^\dagger(x_i^\dagger)) + h'b$. Based on the cost difference bound in (28) and cost assumption in 3.2, we can get a sufficient condition of the anytime competitive constraint as

$$\sum_{i=1}^{h'} \sum_{j=1}^{i} q_{j,h}\|a_j - \pi^\dagger(x_j)\| \leq h'(\lambda\epsilon + b). \tag{29}$$

Since $\sum_{i=1}^{h'} \sum_{j=1}^{i} q_{j,h}\|a_j - \pi^\dagger(x_j)\| \leq \sum_{j=1}^{h'} \Gamma_{j,j}\|a_j - \pi^\dagger(x_j)\|$, this proves Proposition 4.1 for round $h = 1$.

With the cost feedback $\{c_i\}_{i=1}^{h-1}$ collected at round $h$, we can get a lower bound of the prior cost. Based on the cost difference bound in (28) and cost assumption in 3.2 for $i = 1, \cdots, h - 1$ as

$$c_i(x_i^\dagger, \pi^\dagger(x_i^\dagger)) \geq \hat{c}_i^\dagger = \max\left\{ \epsilon, c_i(x_i, a_i) - \sum_{j=1}^{i} q_{j,i}d_j, \right\} \tag{30}$$

where $d_j = \|a_j - \pi^\dagger(x_j)\|, \forall j \in [H]$. Thus, at round $h$, with true cost feedback from the first round to the $(h - 1)$-th round, we get the sufficient condition for the anytime competitive constraint of any $h' \geq h$ as

$$\sum_{i=1}^{h-1} c_i(x_i, a_i) + \sum_{i=h}^{h'} \sum_{j=1}^{i} q_{i,j}\|a_j - \pi^\dagger(x_j)\| \leq (1+\lambda)\sum_{i=1}^{h-1} \max\left\{ \epsilon, c_i - \sum_{j=1}^{i} q_{j,i}d_j \right\} + (h'-h+1)(\lambda\epsilon+b). \tag{31}$$

Recognizing that

$$\sum_{i=h}^{h'} \sum_{j=1}^{i} q_{i,j}\|a_j - \pi^\dagger(x_j)\| = \sum_{j=1}^{h-1} \sum_{i=h}^{h'} q_{j,i}\|a_j - \pi^\dagger(x_j)\| + \sum_{j=h}^{h'} \sum_{i=j}^{h'} q_{j,i}\|a_j - \pi^\dagger(x_j)\|$$
$$\leq \sum_{j=1}^{h-1} \Gamma_{j,h}\|a_j - \pi^\dagger(x_j)\| + \sum_{j=h}^{h'} \Gamma_{j,j}\|a_j - \pi^\dagger(x_j)\| \tag{32}$$

Thus, at round $h$, a sufficient condition for the anytime competitive constraint at round $h'$ can be calculated as

$$\sum_{j=h}^{h'} \Gamma_{j,j}\|a_j - \pi^\dagger(x_j)\| \leq \sum_{i=1}^{h-1} \left( (1+\lambda)\hat{c}_i^\dagger - c_i - \Gamma_{i,h}d_i \right) + (h'-h+1)(\lambda\epsilon+b), \tag{33}$$

which proves Proposition 4.1. $\qquad\square$

## C.2 Proof of Corollary 4.2

**Corollary** 4.2. *At round 1, we initialize the allowed deviation as $D_1 = \lambda\epsilon + b$. At round $h, h > 1$, the allowed deviation is updated as*

$$D_h = \max\{D_{h-1} + \lambda\epsilon + b - \Gamma_{h-1,h-1}d_{h-1}, \ R_{h-1} + \lambda\epsilon + b\}$$

*where $R_{h-1} = \sum_{i=1}^{h-1}\left((1+\lambda)\hat{c}_i^\dagger - c_i - \Gamma_{i,h}d_i\right)$ with notations defined in Proposition 4.1. The $(\lambda, b)-$anytime competitiveness in Definition 3.1 are satisfied if it holds at each round $h$ that $\Gamma_{h,h}\|a_h - \pi^\dagger(x_h)\| \le D_h$.*

*Proof.* We prove by induction. At the first round, by Proposition 4.1, the sufficient condition for the anytime competitive constraint at round $h' \ge 1$ is

$$\sum_{j=1}^{h'}\Gamma_{j,j}\|a_j - \pi^\dagger(x_j)\| \le D_1 + (h'-1)(\lambda\epsilon + b), \ \forall h' = 1, \cdots, H. \tag{34}$$

Thus, with $D_1 = \lambda\epsilon + b$, $\Gamma_{1,1}\|a_1 - \pi^\dagger(x_1)\| \le D_1$ satisfies (34) for $h' = 1$.

Assume that at round $h-1$, the sufficient condition for anytime competitive constraint at round $h' \ge h-1$ is

$$\sum_{j=h-1}^{h'}\Gamma_{j,j}\|a_j - \pi^\dagger(x_j)\| \le D_{h-1} + (h'-h+1)(\lambda\epsilon + b), \ \forall h' = h-1, \cdots, H. \tag{35}$$

Thus, at round $h$, a sufficient condition for anytime competitive constraint at round $h' \ge h$ is

$$\sum_{j=h}^{h'}\Gamma_{j,j}\|a_j - \pi^\dagger(x_j)\| \le D_{h-1} + (h'-h+1)(\lambda\epsilon + b) - \Gamma_{h-1,h-1}d_{h-1}, \ \forall h' = h, \cdots, H. \tag{36}$$

Also, by applying Proposition 4.1 for round $h$, we get another sufficient condition for anytime competitive constraint at round $h' \ge h$ as

$$\sum_{j=h}^{h'}\Gamma_{j,j}\|a_j - \pi^\dagger(x_j)\| \le R_{h-1} + (h'-h+1)(\lambda\epsilon + b), \ \forall h' = h, \cdots, H. \tag{37}$$

Choose the maximum bound in the two sufficient conditions, we get the sufficient condition for anytime competitive constraint at round $h' \ge h$ as

$$\sum_{j=h}^{h'}\Gamma_{j,j}\|a_j - \pi^\dagger(x_j)\| \le D_h + (h'-h)(\lambda\epsilon + b), \ \forall h' = h, \cdots, H. \tag{38}$$

Therefore, for any $h \in [H]$, the sufficient condition for anytime constraint at round $h' \ge h$ is (38). By choosing $a_h$ such that $\Gamma_{h,h}\|a_h - \pi^\dagger(x_h)\| \le D_h$ at each round, (38) is satisfied for round $h' = h \in [H]$ and so the anytime competitive constraints are satisfied for all rounds. $\square$

## D Proofs in Section 5

### D.1 Proof of Theorem 5.1

**Theorem 5.1.** *Assume that the optimal-unconstrained policy $\pi^*$ has a value function $Q_h^{\pi^*}(x, a)$ which is $L_{Q,h}$-Lipschitz continuous with respect to the action $a$ for all $x$. The regret between the optimal ACD policy $\pi^\circ$ that satisfies $(\lambda, b)-$anytime competitiveness and the optimal-unconstrained policy $\pi^*$ is bounded as*

$$\mathbb{E}_{x_1}\left[V_1^{\pi^*}(x_1) - V_1^{\pi^\circ}(x_1)\right] \le \mathbb{E}_{y_{1:H}}\left\{\sum_{h=1}^{H} L_{Q,h}\left[\eta - \frac{1}{\Gamma_{h,h}}(\lambda\epsilon + b + \Delta G_h)\right]^+\right\}, \tag{39}$$

where $\eta = \sup_{x \in \mathcal{X}} \|\pi^*(x) - \pi^+(x))\|$ is the maximum action discrepancy between the policy prior $\pi^\dagger$ and optimal-unconstrained policy $\pi^*$; $\Gamma_{h,h}$ is defined in Proposition 4.1; $\Delta G_h = [R_{h-1}]^+$ is the gain of the allowed deviation by applying Proposition 4.1 at round $h$.

We first define a projected policy based on the optimal-unconstrained policy $\pi^*(s) = \pi^*(x)$ and augmented state $s$ as

$$\pi_h^\perp(s) = \arg \min_{a \in \mathcal{A}_h(D_h)} \|a - \pi_h^*(s)\|. \tag{40}$$

In the next Lemma, we decompose the concerned regret based on the newly-defined policy $\pi_h^\perp$.

**Lemma D.1.** *If $Q_h^{\pi^*}$ is $L_{Q,h}-$ Lipschitz continuous with respect to the action, then the regret between $\pi^\circ$ and $\pi^*$ can be bounded as*

$$\mathbb{E}_{x_1} \left[ V_1^{\pi^\circ}(s_1) - V_1^{\pi^*}(s_1) \right] \leq \mathbb{E}_{y_{1:H}} \left[ \sum_{h=1}^H L_{Q,h} \|\pi_h^\perp(s_h^\perp) - \pi_h^*(s_h^\perp)\| \right]. \tag{41}$$

*Proof.* Let $\xi_h(s) = Q_h^{\pi^*}(x, \pi_h^\perp(s)) - Q_h^{\pi^*}(x, \pi_h^*(s))$. For any round $h$ and any augmented state $s$ obtained by ACD policy, we can bound the value difference as

$$
\begin{aligned}
V_h^{\pi^\circ}(s) - V_h^{\pi^*}(s) &= \tilde{Q}_h^{\tilde{\pi}^*}(s, \tilde{\pi}^*(s)) - Q_h^{\pi^*}(s, \pi^*(s)) \\
&\leq \tilde{Q}_h^{\tilde{\pi}^*}(s, \pi^\perp(s)) - Q_h^{\pi^*}(s, \pi^*(s)) \\
&= Q_h^{\pi^\circ}(s, \pi^\perp(s)) - Q_h^{\pi^*}(s, \pi^\perp(s)) + Q_h^{\pi^*}(s, \pi^\perp(s)) - Q_h^{\pi^*}(s, \pi^*(s)) \\
&= \mathbb{E}_{s_{h+1}} \left[ V_{h+1}^{\pi^\circ}(s_{h+1}) - V_{h+1}^{\pi^*}(s_{h+1}) \mid s, \pi^\perp(s) \right] + \xi_h(s),
\end{aligned} \tag{42}
$$

where the first equality holds since $\pi^\circ$ is the ACD policy based on ML policy $\tilde{\pi}^*(s)$, the inequality holds since $\tilde{\pi}^*$ optimizes $\tilde{Q}_h^{\tilde{\pi}^*}$, the third equality holds since $\pi^\circ$ is the optimal ACD policy with $\tilde{\pi}^*$ as the ML model, and the last equality holds by the definition of $Q_h^\pi$.

Iteratively applying (42), we get

$$
\begin{aligned}
&V_1^{\pi^\circ}(s_1) - V_1^{\pi^*}(s_1) \\
&\leq \left( \prod_{h=1}^H \mathbb{E}_{s_{h+1}|s_h, \pi^\perp(s_h)} \right) \left[ V_{H+1}^{\pi^\circ}(s_{H+1}) - V_{H+1}^{\pi^*}(s_{H+1}) \right] + \sum_{h=1}^H \left( \prod_{i=1}^{h-1} \mathbb{E}_{s_{i+1}|s_i, \pi^\perp(s_i)} \right) [\eta_h(s_h)] \\
&= \mathbb{E}_{y_{1:H}} \left[ \sum_{h=1}^H \eta_h(s_h^\perp) \right] \leq \mathbb{E}_{y_{1:H}} \left[ \sum_{h=1}^H L_{Q,h} \|\pi_h^\perp(s_h^\perp) - \pi_h^*(s_h^\perp)\| \right],
\end{aligned} \tag{43}
$$

where $s_h^\perp$ is the state generated by policy $\pi^\perp$, the last equality holds since $V_{H+1}^\pi = 0$, and the last inequality holds by the Lipschitz continuity of $Q_h^{\pi^*}$. $\square$

**Lemma D.2.** *Given any $s_h$ generated by policy $\pi^\perp$, the action difference between $\pi^\perp$ and $\pi^*$ is bounded as*

$$\|\pi_h^\perp(s_h) - \pi_h^*(s_h)\| \leq \left[ \eta - \frac{1}{\Gamma_{h,h}} (\lambda \epsilon + b + \Delta G_h) \right]^+, \tag{44}$$

*where $\Delta G_h = [R_{h-1}]^+ \geq 0$.*

*Proof.* Since $\pi_h^\perp$ is the projection of $\pi_h^*$ into the action norm ball $\mathcal{A}_h(D_h) = \left\{ a \mid \Gamma_{h,h} \|a - \pi^\dagger(x_h)\| \leq D_h \right\}$, we have

$$
\begin{aligned}
\|\pi_h^\perp(s_h) - \pi_h^*(s_h)\| &= \left[ \|\pi^\dagger(x_h) - \pi_h^*(x_h)\| - \frac{D_h}{\Gamma_{h,h}} \right]^+ \\
&\leq \left[ \eta - \frac{D_h}{\Gamma_{h,h}} \right]^+,
\end{aligned} \tag{45}
$$

where the last inequality holds by the definition of $\eta$.

Since $D_h \geq \Gamma_{h,h} d_h$ for any $h \in [H]$, we have $D_h \geq \lambda \epsilon + b + [R_{h-1}]^+ = \lambda \epsilon + b + \Delta G$. Thus completed the proof. $\square$

**Proof of Theorem 5.1**

*Proof.* By Lemma D.1 and Lemma D.2, we have

$$\mathbb{E}_{x_1}\left[V_1^{\pi^*}(x_1) - V_1^{\pi^\circ}(x_1)\right] = \mathbb{E}_{x_1}\left[V_1^{\pi^*}(s_1) - V_1^{\pi^\circ}(s_1)\right]$$

$$\leq \mathbb{E}_{y_{1:H}}\left[\sum_{h=1}^{H} L_{Q,h}\|\pi_h^\perp(s_h^\perp) - \pi_h^*(s_h^\perp)\|\right] \tag{46}$$

$$\leq \mathbb{E}_{y_{1:H}}\left[\sum_{h=1}^{H} L_{Q,h}\left[\eta - \frac{1}{\Gamma_{h,h}}\left(\lambda\epsilon + b + \Delta G_h\right)\right]^+\right],$$

where $\Delta G_h = [R_{h-1}]^+$. Thus completes the proof. $\qquad\square$

## D.2 Proof of Theorem 5.2

**Theorem 5.2.** *Assume that the value function is bounded by $\bar{V}$. Denote a set of function as*

$$\mathcal{Q} = \{q \mid \exists g \in \mathcal{G}, \forall (s,a,v) \in \mathcal{S} \times \mathcal{A} \times \mathcal{V}, q(s,a,v) = \mathbb{E}_{f \sim g}\left[v(s') \mid s,a\right]\}. \tag{47}$$

*If $\beta_k = 2(\bar{V}H)^2 \log\left(\frac{2\mathcal{N}(\mathcal{Q},\alpha,\|\cdot\|_\infty)}{\delta}\right) + C\bar{V}H$ with $\alpha = 1/(KH\log(KH/\delta))$, $C$ being a constant, and $\mathcal{N}(\mathcal{Q},\alpha,\|\cdot\|_\infty)$ being the covering number of $\mathcal{Q}$, with probability at least $1 - \delta$, the pseudo regret of Algorithm 2 is bounded as*

$$\mathrm{PReg}(K) \leq 1 + d_{\mathcal{Q}}H\bar{V} + 4\sqrt{d_{\mathcal{Q}}\beta_K KH} + H\sqrt{2KH\log(1/\delta)}, \tag{48}$$

*where $d_{\mathcal{Q}} = \dim_E(\mathcal{Q}, \frac{1}{KH})$ is the Eluder dimension of $\mathcal{Q}$ defined in [50].*

**Lemma D.3.** *Assume that $g \in \mathcal{G}_k$, the difference between the reward of the optimal ACD policy $\pi^\circ$ and the reward of policy $\pi^k$ is bounded as*

$$V_1^{\pi^\circ}(s_1) - V_1^{\pi^k}(s_1) \leq \sup_{\tilde{g}\in\mathcal{G}_t} \sum_{h=1}^{H-1} \mathbb{E}_{\tilde{g}-g}\left[\tilde{V}_{h+1}^k(s_{h+1}^k) \mid s_h^k, a_h^k\right] + \sum_{h=1}^{H-1} \xi_{h+1,k}, \tag{49}$$

*where $\xi_{h+1,t} = \mathbb{E}_g\left[\tilde{V}_{h+1}^k(s_{h+1}) - V_{h+1}^{\pi^k}(s_{h+1})\right] - \left[\tilde{V}_{h+1}^k(s_{h+1}) - V_{h+1}^{\pi^k}(s_{h+1})\right].$*

*Proof.* Since the true transition model $g \in \mathcal{G}_k$, we have $V_1^{\pi^\circ}(s_1^k) \leq \tilde{V}_1^k(s_1^k)$, and so

$$V_1^{\pi^\circ}(s_1^k) - V_1^{\pi^k}(s_1^k) \leq \tilde{V}_1^k(s_1^k) - V_1^{\pi^k}(s_1^k). \tag{50}$$

At round $h$, we have

$$\tilde{V}_h^k(s_h^k) - V_h^{\pi^k}(s_h^k)$$

$$= \left(r(s_h^k, a_h^k) + \mathbb{E}_{g^k}\left[\tilde{V}_{h+1}^k(s_{h+1}) \mid s_h^k, a_h^k\right]\right) - \left(r(s_h^k, a_h^k) + \mathbb{E}_g\left[V_{h+1}^{\pi^k}(s_{h+1}) \mid s_h^k, a_h^k\right]\right)$$

$$= \sum_{f\in\mathcal{F}} \tilde{V}_{h+1}^k(f(x_h, a_h), D_{h+1})g^k(f) - \sum_{f\in\mathcal{F}} V_{h+1}^{\pi^k}(f(x_h, a_h), D_{h+1})g(f)$$

$$= \sum_{f\in\mathcal{F}} \tilde{V}_{h+1}^k(f(x_h, a_h), D_{h+1})\left(g^k(f) - g(f)\right) \tag{51}$$

$$+ \sum_{f\in\mathcal{F}} \left(\tilde{V}_{h+1}^k(f(x_h, a_h), D_{h+1}) - V_{h+1}^{\pi^k}(f(x_h, a_h), D_{h+1})\right)g(f)$$

$$= \mathbb{E}_{g^k-g}\left[\tilde{V}_{h+1}^k(s_{h+1}^k) \mid s_h^k, a_h^k\right] + \mathbb{E}_g\left[\tilde{V}_{h+1}^k(s_{h+1}) - V_{h+1}^{\pi^k}(s_{h+1}) \mid s_h^k, a_h^k\right]$$

Let $\xi_{h+1,t} = \mathbb{E}_g\left[\tilde{V}_{h+1}^k(s_{h+1}) - V_{h+1}^{\pi^k}(s_{h+1})\right] - \left[\tilde{V}_{h+1}^k(s_{h+1}) - V_{h+1}^{\pi^k}(s_{h+1})\right].$ By the fact that $V_{H+1} = 0$ and summing $\left[\tilde{V}_h^k(s_h^k) - V_h^{\pi^k}(s_h^k)\right] - \left[\tilde{V}_{h+1}^k(s_{h+1}) - V_{h+1}^{\pi^k}(s_{h+1})\right] =$

$\mathbb{E}_{g^k-g}\left[\tilde{V}_{h+1}^k(s_{h+1}^k) \mid s_h^k, a_h^k\right] + \xi_{h+1,t}$ from $h = 1$ to $H$, we have

$$
\begin{aligned}
&\tilde{V}_1^k(s_1^k) - V_1^{\pi^k}(s_1^k) \\
&= \sum_{h=1}^{H-1} \mathbb{E}_{g^k-g}\left[\tilde{V}_{h+1}^k(s_{h+1}^k) \mid s_h^k, a_h^k\right] + \sum_{h=1}^{H-1} \xi_{h+1,k} \\
&\leq \sup_{\tilde{g} \in \mathcal{G}_t} \sum_{h=1}^{H-1} \mathbb{E}_{\tilde{g}-g}\left[\tilde{V}_{h+1}^k(s_{h+1}^k) \mid s_h^k, a_h^k\right] + \sum_{h=1}^{H-1} \xi_{h+1,k}
\end{aligned}
\tag{52}
$$

$\square$

**Lemma D.4** ([6]). *Let* $\beta_k = 2H^2 \log\left(\frac{2\mathcal{N}(\mathcal{Q},\alpha,\|\cdot\|_\infty)}{\delta}\right) + 2H(kH-1)\alpha\left\{2 + \sqrt{\log(\frac{4kH(kH-1)}{\delta})}\right\}$
*with* $\alpha > 0$. *Then with probability* $1 - \delta, \delta \in (0,1)$, *we have* $g \in \mathcal{G}_k$ *for any* $k \geq 1$.

**Lemma D.5** ([50]). *Let* $d_\mathcal{Q} = \dim_E(\mathcal{Q}, \frac{1}{KH})$ *be the Eluder dimension of the function set* $\mathcal{Q}$ *defined in* (14). *When* $g \in \cap_{k \in [K]}\mathcal{G}_k$, *the cumulative value estimation error is bounded as*

$$
\sup_{\tilde{g} \in \mathcal{G}_t} \sum_{h=1}^{H-1} \mathbb{E}_{\tilde{g}-g}\left[\tilde{V}_{h+1}^k(s_{h+1}^k) \mid s_h^k, a_h^k\right] \leq 1 + Hd_\mathcal{Q} + 4\sqrt{d_\mathcal{Q}\beta_K KH}.
\tag{53}
$$

**Proof of Theorem 5.2**

*Proof.* By choosing $\alpha = 1/(KH \log(KH/\delta))$ and the assumption that $V$ is upper bounded by $\bar{V}$, the $\beta_k$ in Lemma D.4 is expressed as $\beta_k = 2(\bar{V}H)^2 \log\left(\frac{2\mathcal{N}(\mathcal{Q},\alpha,\|\cdot\|_\infty)}{\delta}\right) + C\bar{V}H$.

Since $\xi_{2,1}, \cdots, \xi_{H,1}, \cdots, \xi_{2,K}, \cdots, \xi_{H,K}$ is a martingale sequence, with probability at $1 - \delta$, the sampling error is bounded as

$$
\sum_{t=1}^{T}\sum_{h=1}^{H} \xi_{h=1}^{H} \leq T\sqrt{2TH \log(1/\delta)}.
\tag{54}
$$

By Lemma D.5 and union bound, we have with probability with $1 - \delta, (\delta \in (0,1))$,

$$
V_1^{\pi^\circ}(s_1) - V_1^{\pi^k}(s_1) \leq 1 + Hd_\mathcal{Q}\bar{V} + 4\sqrt{d_\mathcal{Q}\beta_K KH} + T\sqrt{2TH \log(2/\delta)}.
\tag{55}
$$

$\square$