# OpenReview forum: "Anytime-Competitive Reinforcement Learning with Policy Prior"
_NeurIPS.cc/2023/Conference — NeurIPS 2023 poster_

### Official Review · Reviewer_LDWt · 2023-06-14

**Soundness:** 3 good
**Presentation:** 3 good
**Contribution:** 3 good
**Rating:** 6
**Confidence:** 4

**Summary:**

This paper addresses the problem of anytime constrained RL. Contrary to traditional constrained RL where the constraints are expressed as expected costs, this work looks at a stronger criteria that the cost should be bounded for any episode. The cost is defined relative to the performance of an existing safe policy deployed in the environment of interest.

An algorithm (ACD) is derived which consists in deriving a safe action set by computing the maximum allowed deviation from the prior policy. The computation relies on the knowledge of Lipschitz constants characterizing the transition model, the cost variation, and the policy prior. This algorithm allows to take safe actions but does not address optimizing the return.

ACD is extended to ACRL which adds the learning and optimization parts to maximize the return while taking safe actions. ACRL is a model based algorithm. An analysis of the regret of the algorithm is provided to analyze the regret from the anytime constraint and from the learning part.


**Strengths:**

The introduced framework of A-CMDP is very sound and fits many practical applications. The main assumption of the work that there exist a policy prior is a strong one but I believe it applies to a lot of problems and is quite relevant. The associated definition of regret makes sense, although the formalism of the A-CMDP is sometimes unclear.

The article is fairly well written. All propositions and assumptions are followed by an intuitive explanation and they all seem sound.

The discussion of the related work and their relation to what is being presented is clear and seems to cover most of the relevant topics.

The structure of deriving first ACD and ACRL is useful to follow the work and the algorithms make sense given the proposed assumptions.


**Weaknesses:**

The authors did not include the experiments in the main body of the paper. The experiments show that the algorithm is better in terms of worst case cost, but does not provide the best regret per episode (expected since stronger constraints). The algorithm is evaluated on only one problem, it would have been useful to test it on at least one other problem to show its value, maybe some classical benchmark from the safe RL literature. An ablation study of showing the performance of the model-based approach without the constraints would have been useful as well. Adding the experiments to the main body seems necessary to me.

Although the intuitive explanation of the A-CMDP is clear, there are some unclarities in the mathematical definition of the A-CMDP (see questions).

The main assumptions of the article require the cost functions, transition functions, and prior policy to have some sort of Lipschitz continuity property with known constants. At the same time, a main advantage of the proposed algorithm is that it does not require to know much of the dynamics, it seems a bit contradictory. The author provide some arguments with support from related work but I am still curious whether a precise knowledge of this constant is important or not.

In ACRL, the state augmentation part increasing significantly the size of state by adding a dependency on the history. This approach raises scalability issues which are not discussed in the paper.


**Questions:**

- I am not sure I understand why the notion of model (y) is needed in the definition the A-CMDP, as opposed to the notion of trajectories more common in the RL literature?

- Why is the value definition an expectation over the first state $x_1$? Shouldn’t it be an expectation over trajectories like in regular MDPs?

- How does one find the necessary Lipschitz constants in a problem where the dynamics is unknown? Is it important to have accurate values?

- In proposition 4.1, how can the cost of the policy prior be known if it is not executed?


**Limitations:**

The authors are explicitly mentioning the limitations at the end of the paper. However, I find the discussion somewhat contradicting. One of the main related works rely on knowledge of the dynamics and this work relies on knowledge of the Lipschitz constants and some information about the policy prior. Is it really much simpler to get this information compared to an approximate dynamics model?

---

> ### Author Rebuttal · Authors · 2023-08-04
>
> We appreciate your comments and are happy to address them as below.
>
> `Experiments.`\
> Thank you for your suggestions, we will move some important experiment results to the main body of the paper.
>
> Since the motivation is from mission-critical applications, we will add another experiment on real-time voltage control [1] where the Lyapunov-based policy is used as the policy prior.
>
> `Notion y in the definition the A-CMDP?`\
> A sampled sequence of models can be seen as a part of a sampled trajectory. We will change expression as $\sum_{i=1}^hc_i(x_i,a_i)\leq (1+\lambda) \sum_{i=1}^hc_i(x_i^{\dagger},a_i^{\dagger})+hb$ to avoid any misunderstanding and be more consistent with the RL literature.
>
> `Value definition in the objective (1)`\
> The value function $V(x_h)=\mathbb{E}[\sum_{i=h}^Hc_i(x_i,a_i)\mid x_h]$ is an expectation of reward over the trajectories conditioned on a state $x_h$. We add an expectation $E_{x_1}$ over the initial state $x_1$ on $V(x_1)$ in (1) to represent the expected reward over the whole trajectory.
>
> `Is the knowledge about Lipschitz continuity and telescoping property important? How to get the knowledge of Lipschitz constants and telescoping parameters? Is it easier than approximating a dynamic model?`\
> We consider any-time constraints which are required to be satisfied for any round in any episode.  Different from standard constrained MDPs where constrained are satisfied in expectation or with high probability, the any-time constraints cannot be guaranteed without further knowledge on the environment. In this work, we find that the any-time constraints can be theoretically guaranteed with the knowledge of Lipschitz constants and telescoping parameters because they indicate how sensitive the system is given a small perturbation of states or actions.  Given this intuition, knowing the upper bounds of the Lipschitz constants and telescoping parameters is enough to guarantee the constraints by our algorithm. In spite of this,  Theorem 5.1 shows that using larger Lipschitz constants or perturbation function in ACD can cause larger reward gap between the optimal ACD policy and the optimal unconstrained policy, resulting in a larger bound of the reward regret. This is because smaller safe action sets are obtained under this case, causing less flexibility to optimize the reward.
>
> Knowing the Lipschitz constants and telescoping parameters are far from approximating a dynamic model. There exists a set of transition functions that satisfy the Lipschitz assumption and the telescoping property given fixed Lipschitz constants and telescoping parameters. Thus, assuming the knowledge does not contradict with the role of RL in learning dynamic models.
>
> For mission-critical applications [1,5,6], it is much easier to get the needed information based on some prior knowledge. The Lipschitz constants of cost and transition functions can be estimated based on the model information. For telescoping parameters, we can first estimate the Lipschitz constant of the policy prior as $L_{\pi^{\dagger}}$ given that we have full access to the policy prior. Then the telescoping function can be simply calculated as $p(h_2-h_1)=(L_f(1+L_{\pi^{\dagger}}))^{h_2-h_1}$ where $L_f$ is the Lipschitz constant of the transition function. For some specific priors and dynamics (e.g. priors in nonlinear stability analysis [7]), tighter telescoping functions can be calculated.
>
> `In proposition 4.1, how can the cost of the policy prior be known if it is not executed?`\
> In proposition 4.1, the cost of the policy prior $c^{\dagger}$ is never needed. Instead, we derive a lower bound of the cost of the policy prior, i.e.  $\hat{c}^{\dagger}$ based on the known parameters to evaluate inequality (4).\
>
>
> `The scalability issue in ACRL`\
> The augmented state in Line 254-258 includes the history costs and action differences which are needed to calculate the safe action set for each round. The dimension of the augmented state increases with the length of the horizon $H$, resulting a scalability issue. The salability issue also exits in other RL works [2,3].  in this paper, we focus on the constraint guarantee and its impact on the regret, but in practice, tractable methods can be designed through feature aggregation [2] or PODMP [4].
>
>
>
> **References**\
> [1] Shi, Y., Qu, G., Low, S., Anandkumar, A. and Wierman, A., 2022, June. Stability constrained reinforcement learning for real-time voltage control. In 2022 American Control Conference (ACC) (pp. 2715-2721). IEEE.\
> [2] Tennenholtz, Guy, et al. "Reinforcement Learning with History-Dependent Dynamic Contexts." ICML 2023.\
> [3] Chen, Xiaoyu, et al. "An adaptive deep rl method for non-stationary environments with piecewise stable context." NeurIPS 2022.\
> [4] Xiong, Yi, et al. "Sublinear regret for learning pomdps." Production and Operations Management 31.9 (2022): 3491-3504.\
> [5] Luo, Jerry, et al. "Controlling commercial cooling systems using reinforcement learning." arXiv preprint arXiv:2211.07357 (2022).\
> [6] Radovanović, Ana, et al. "Carbon-aware computing for datacenters." IEEE Transactions on Power Systems 38.2 (2022): 1270-1280.\
> [7] Tsukamoto, Hiroyasu, Soon-Jo Chung, and Jean-Jaques E. Slotine. "Contraction theory for nonlinear stability analysis and learning-based control: A tutorial overview." Annual Reviews in Control 52 (2021): 135-169.

---

> > ### Comment · Reviewer_LDWt · 2023-08-10
> > **Thank you for the clarifications, please add the experiments to the main text**
> >
> > I thank the author for answering all my questions and comments, I acknowledge reading the rebuttal.
> >
> > I am willing to keep my recommendation for acceptance, since they author plan to add the experiment results in the main body of the paper. I would also appreciate it if they can make the scalability issue more obvious in the text.
> >
> > Being a bit more picky on the value definition, I think it is missing the dependency on the policy to be correct. It is conditioned on the starting state but should be also conditioned on the actions following a given policy.

---

> > > ### Author Response · Authors · 2023-08-10
> > > **Responses to further comments.**
> > >
> > > We appreciate your valuable suggestions and willingness to keep your recommendation for acceptance!
> > >
> > > We will add the experiment results in the main body of the paper and discuss the scalability issue obviously and clearly in the paper .
> > >
> > > Regarding the value definition, we will use $V_h^{\pi}(x_h)$ to represent the expected total reward starting from round $h$ to $H$ conditioned on a policy $\pi$ and a starting state $x_h$.
> > >
> > > We are happy to address any further comments you may have!

---

> > > > ### Comment · Reviewer_LDWt · 2023-08-15
> > > >
> > > > I updated the grade from 5 to 6.

---

### Official Review · Reviewer_MVJu · 2023-06-20

**Soundness:** 3 good
**Presentation:** 2 fair
**Contribution:** 2 fair
**Rating:** 5
**Confidence:** 4

**Summary:**

Existing work on Constrained Markov Decision Processes (CMDPs) typically formulate a problem where the expected cumulative reward
is maximized under a constraint where the expected cost over random dynamics is less than some threshold. Such existing work has a drawback that the safety constraint is violated in a specific episode. Hence, the authors study a problem named Anytime-CMDP (A-CMDP). The objective of the authors is to optimize the reward value function while guaranteeing the satisfaction of the safety constraint in each round of any episode. Under several assumptions (e.g., telescoping policy prior, Lipschitz continuity), the authors propose a new algorithm, called Anytime-Constrained Reinforcement Learning (ACRL) and provided theoretical results on constraint satisfaction and regret. The authors conduct experiments on the application of carbon-intelligent computing.

**Strengths:**

- Important problem settings. The drawbacks of the typical CMDPs have been discussed in recent years, so I think this paper will attract much attention within safe RL research community.

- Good theoretical results. Though there are many assumptions, the obtained theoretical results are indeed good. Given the difficulty of the problem (and the theoretical results the authors try to provide), the number and strength of the assumptions are somewhat reasonable.

**Weaknesses:**

- There are several important missing citations and comparisons. Especially, Ref-1 is very relevant to this paper, which try to guarantee the satisfaction of the safety constraint with probability of 1. This paper is exactly the same motivation to Ref-1 though the approaches are quite different. Though I understand there are pros and cons as follows, it is extremely important to discuss the differences between Ref-1 and this paper.

    ```
    [Ref-1]
        + High performance in complicated environments (e.g., Safety-Gym)
        + High affinity with deep RL
        + Assumptions are weak
        - No theoretical guarantee
    [This paper]
        + Theoretical guarantee
        - High affinity with deep RL
        - Assumptions are strong
    ```

- Also, anytime constraints have been also studied in several literature. For example, [Ref-2] and [Ref-3] formulate their safety constraint as an instantaneous one $c(s_t, a_t) \le d$ and propose algorithms to guarantee its satisfaction with high probability (though it is not probably of 1). It seems quite miss-leading that Table 1 show that ACRL is the first and only work to deal with any-time constraint.

- Regarding experiments, the benchmark problem used in this paper is not popular and it would be better to evaluate their method in standard ones (e.g., Safety-Gym). Especially, if I understand correctly, the SOTA algorithm in the problem settings the authors address is Saute-RL in [Ref-1] and the I consider that there should be some comparison between ACRL and Saute-RL.

- There are so many strong assumptions, which makes us to follow this paper. I guess it is partly because the authors try to derive regret bound as well as guarantee safety. I wonder what is the minimal set of assumptions for guaranteeing safety. It may be better to focus on guaranteeing safety with a small number of assumptions, which make this paper easy to follow and the contribution clearer.

- References

    - [Ref-1] Sootla, Aivar, et al. "Sauté RL: Almost surely safe reinforcement learning using state augmentation." International Conference on Machine Learning. PMLR, 2022.

    - [Ref-2] Wachi, Akifumi, and Yanan Sui. "Safe reinforcement learning in constrained Markov decision processes." International Conference on Machine Learning. PMLR, 2020.

    - [Ref-3] Amani, Sanae, Christos Thrampoulidis, and Lin Yang. "Safe reinforcement learning with linear function approximation." International Conference on Machine Learning. PMLR, 2021.

**Questions:**

[Q1] Is there any reason why standard safe RL benchmark problems (e.g., Safety Gym) were not used?

[Q2] What is the minimal set of assumptions that are needed for guaranteeing safety?

**Limitations:**

Limitations have been adequately addressed.

---

> ### Author Rebuttal · Authors · 2023-08-02
>
> We appreciate your comments and are happy to address them as below.
>
> `The differences between Ref-1 and this paper.` \
> Ref-1 solves the MDP with almost surely safety constraint by introducing Sauté MDP where constraints are incorporated into the reshaped objective. The approach is verified to have a high empirical performance, but there is no theoretical guarantee for constraint satisfaction, as is summarized by the reviewer. Comparably, we derive the sufficient conditions of the constraint satisfaction in Proposition 4.1 and Corollary 4.2, and propose a projection-based policy with theoretical guarantee for the any-time constraint satisfaction. The regret analysis shows the impact of the strict constraint satisfaction on reward optimization. We will discuss the differences in the future versions.
>
> We'd like to highlight that the theoretical guarantee for the any-time constraints is essential for safety-critical applications with examples in Section 3.2.  In the application of cooling control in datacenters [6], for example, even one violation of the safety constraint can cause unaffordable loss.  The lack of theoretical constraint guarantee hinders the deployment of deep policies in real safety-critical systems. Thus we argue that although with some reasonable assumptions (explained in the next answer), the proposed RL algorithm with theoretically guaranteed any-time constraints is really needed.
>
> `Reasons for the assumptions.`\
> The minimal set of assumptions to guarantee safety are listed in Assumption 3.2 and 3.4 which include the known cost lower bound, the Lipschitz cost and transition functions with known Lipschitz constants, and the existence of a policy prior satisfying telescoping property.  The cost lower bound can be as small as zero. The non-negative cost is also assumed in [Ref-1].  The Lipschitz cost functions are assumed in other MDPs [8]. The Lipschitz transition functions are common in sequential decision-making problems [3,4,9].  In mission-critical applications such as voltage control [1] and cooling control[2], the Lipschitz constants can be estimated based on prior knowledge. Additionally, the telescoping property holds for many policy priors, e.g. stable policies [5].  It is also proved for model predictive control [4], and it is assumed for perturbation analysis in [3].
>
> To the best of our knowledge, our work presents the first RL algorithm to guarantee the any-time constraint under these assumptions which are reasonable for many mission-critical applications. However, for the future work, it is definitely interesting to study RL that guarantee the constraints under relaxed assumptions to be applied to general scenarios.
>
> `Why not evaluate on standard benchmarks?`\
> The settings and assumptions of ACMDP are motivated from the sequential decision-making problems in mission-critical applications such as datacenters [7], cooling systems [2,6], power systems[1], In these systems, there exist some safe policy priors that have been programmed in the real systems for a long time and/or have verified cost performance. Also, the key system (environment) properties (in Assumption 3.2 and 3.4) are known to the agent. The considered application of carbon-aware computing in datacenters is one of such applications, so we evaluate our method ACRL in this application.
>
> The standard benchmarks generally do not meet these assumptions. The method in SOTA algorithms like Ref-1 applies to the environments with safety budget constraints where the agent has less knowledge about the environment. Since the considered constraints are different, it is not fair to compare our paper with these algorithms.
>
> `The any-time constraint in Table 1.` \
> The "any-time constraint" specifically refers to the any-time cost constraint against a policy prior in Eqn. (1) which is studied by us for the first time. We will change the phrase in Table 1 into any-time constraint against a policy prior to avoid misunderstanding.
>
>
> **References**\
> [1] Shi, Yuanyuan, et al. "Stability constrained reinforcement learning for real-time voltage control." ACC 2022.\
> [2] Luo, Jerry, et al. "Controlling commercial cooling systems using reinforcement learning." arXiv preprint arXiv:2211.07357 (2022).\
> [3] Lin, Yiheng, et al. "Bounded-Regret MPC via Perturbation Analysis: Prediction Error, Constraints, and Nonlinearity." NeurIPS 2022.\
> [4] Lin, Yiheng, et al. "Perturbation-based regret analysis of predictive control in linear time varying systems." NeurIPS 2021.\
> [5] Tsukamoto, Hiroyasu, Soon-Jo Chung, and Jean-Jaques E. Slotine. "Contraction theory for nonlinear stability analysis and learning-based control: A tutorial overview." Annual Reviews in Control 52 (2021): 135-169.\
> [6] Google Deepmind, Safety-first ai for autonomous data centre cooling and industrial control. \
> [7] Radovanović, Ana, et al. "Carbon-aware computing for datacenters." IEEE Transactions on Power Systems 38.2 (2022): 1270-1280.\
> [8] Luo, Fan-Ming, et al. "A survey on model-based reinforcement learning." arXiv preprint arXiv:2206.09328 (2022).\
> [9] Gradu, Paula, John Hallman, and Elad Hazan. "Non-stochastic control with bandit feedback." NeurIPS 2020.

---

> > ### Comment · Reviewer_MVJu · 2023-08-11
> > **Thank you for rebuttals**
> >
> > Thank you for the convincing rebuttals.
> >
> > **The differences between Ref-1 and this paper.** I understand the differences. It would be helpful for the readers to add such discussion. Please include it in the camera-ready version.
> >
> > **Reasons for the assumptions.** Though I still consider it quite strong, I understand the assumptions are indeed necessary for solving safe RL problems.
> >
> > **Why not evaluate on standard benchmarks?** I understand. If the benchmark is not standard one, it is more necessary to release the source-code for reproducibility. Please consider open-sourcing the source code for RL environments and agents.
> >
> > **The any-time constraint in Table 1.** I recommend the authors to improve the Table 1 for avoid over-claiming.
> >
> > Under the condition that the authors improve the paper based on the above comments, I increased the score from 3 to 5.

---

> > > ### Author Response · Authors · 2023-08-11
> > > **Responses to further comments.**
> > >
> > > We appreciate your valuable suggestions and thank you for increasing the score!
> > >
> > > We will comprehensively discuss the differences between our work and Ref-1 in the main paper.
> > >
> > > Furthermore, we will certainly open-source the codes including the RL environments and policy learning as soon as the paper is published.
> > >
> > > The entries in Table 1 will be carefully revised to ensure an accurate representation.
> > >
> > > We are happy to address any further comments you may have!

---

### Official Review · Reviewer_skPV · 2023-07-04

**Soundness:** 3 good
**Presentation:** 3 good
**Contribution:** 3 good
**Rating:** 6
**Confidence:** 4

**Summary:**

The paper studies a novel problem of Anytime-Constrained Markov Decision Process (A-CMDP), where the goal is to optimize the expected reward while guaranteeing that the cumulative cost in each round of any episode does not exceed a relaxed version of the cost of a policy prior. The paper proposes a projection-based algorithm, called Anytime-Constrained Decision-making (ACD), which ensures the anytime cost constraints by projecting the output of a machine learning policy into safe action sets at each round. The paper also develops an efficient model-based reinforcement learning algorithm, called Anytime-Constrained Reinforcement Learning (ACRL), which learns the optimal ML policy for ACD via value iteration and model estimation. The paper provides theoretical analysis on the regret performance of ACRL, showing the trade-off between reward optimization and anytime constraint satisfaction, as well as the effectiveness of policy learning.

**Strengths:**


- The paper addresses an important and novel problem of A-CMDP, which has direct motivations from many mission-critical applications that require strict cost constraints for safety and reliability.
- The paper introduces a novel algorithm, ACD, which can guarantee the anytime constraints given any ML policy by projecting it into safe action sets. The paper also develops a model-based RL algorithm, ACRL, which can learn an optimal ML policy to be used in ACD.
- The paper provides rigorous theoretical analysis on the regret performance of ACRL and shows how it depends on the relaxation parameters and the discrepancy between the policy prior and the optimal-unconstrained policy.
- The paper presents empirical results on an application of resource management for carbon-aware computing, although all of this part is in appendix.

**Weaknesses:**


- The paper makes some strong assumptions on the Lipschitz continuity of the cost and transition functions, and the telescoping property of the policy prior, which may limit its applicability to more general environments. Specifically, the telescoping property of the policy prior is a strong assumption for both the policy and the dynamics. It would be interesting to explore how to relax these assumptions or how to estimate them online without compromising safety.
- The paper also relies on the assumption that the safe policy prior can be queried on any state, even if the state is not explored on the policy prior trajectory. This assumption may not be feasible in real applications.
- The paper does not provide much intuition or discussion on how the relaxation parameters or other parameters would affect the exploration-exploitation trade-off and the robustness of ACRL.

**Questions:**


- How does ACRL scale with large state-action spaces or long horizons? What would be the challenges there and how would the scale affect the convergence?
- What are the computational challenges or bottlenecks in implementing ACRL?
- How sensitive is ACRL to errors or uncertainties in estimating the Lipschitz constants, minimum cost value, or perturbation function?
- In some applications there might be several constraints. How can ACRL be extended to handle multiple costs or multiple constraints?

**Limitations:**

The authors have adequately listed most limitations in their work, such as assuming Lipschitz continuity, telescoping property, and known parameters. Those are reasonable to some extent since as stated by the authors, there is no way to guarantee any time safety constraints without any knowledge of the properties of the dynamics. However, they could also discuss some possible extensions or future directions based on their limitations, such as relaxing some assumptions, incorporating online estimation, exploring more applications, or improving scalability.

---

> ### Author Rebuttal · Authors · 2023-08-04
>
> We appreciate the comments of the reviewer and address them as below.
>
> `The reason that the safe policy prior can be queried on any state`\
> The safe policy prior is a mapping from a state to an action. In the ACD policy, policy prior is queried on true states to get a virtual action (which does not interact with the real world) to construct the safe action set. The queries bring some computation load but they are doable in the real world.
>
> `How do the relaxation parameters or other parameters affect the exploration-exploitation trade-off and the robustness of ACRL?`\
> The relaxation parameters $\lambda$ and $b$ affects the trade-off between the reward maximization and the any-time constraint satisfaction as shown in Theorem 5.1. When $\lambda$  or $b$ becomes larger,  the any-time constraints in (1) are relaxed to have more flexibility to optimize the average reward, so the regret comparing with the optimal unconstrained policy becomes smaller.
>
> The relaxation parameters $\lambda$ and $b$ also have effect on the exploration complexity. With smaller $\lambda$ or $b$, the safe action set in (7) becomes smaller, which results in a smaller action-state space $\mathcal{A}\times \mathcal{S}$ for the new MDP defined at the beginning of Section 4.2. By Theorem 5.2, the regret becomes smaller since the covering number and the Eluder dimension of set $\mathcal{Q}$ is reduced with smaller action-state space.
>
> Another important parameter that affect the exploration-exploitation trade-off is the confidence parameter $\beta_k$ in (10). When $\beta_k$ becomes larger, ACRL explores more, and vice versa. The regret is bounded in Theorem 5.2 with a proper choice of $\beta_k$.
>
> We will add these intuition and discussions in the future version.
>
> `How does ACRL scale with large state-action spaces or long horizons?`\
> The dependence of ACRL regret on the length of horizons $H$ is explicitly shown in (16).  Increasing the size of the state-action space can cause a larger covering number and Eluder dimension of the value function set $\mathcal{Q}$, resulting in a larger regret. The challenges from large state-action space and long horizon also exist in many RL works ([1,2,3]). In this paper, we focus on the constraint guarantee and its impact on the regret, and leave the scalability issue on regret for future works.
>
> `What are the computational challenges or bottlenecks in implementing ACRL?`\
> The computational challenges of ACRL mainly comes from the scaling of state-action space and horizon length. For implementation, the scalability issue can be mitigated by feature aggregation [4].
>
> Also, ACRL needs to query the policy prior and solve a projection into the safe action ball (7) at each round, which brings additional computational complexity. Some convex optimizers [5] can be queried to solve the projections
>
> `How sensitive is ACRL to errors or uncertainties in estimating the Lipschitz constants, minimum cost value, or perturbation function?`\
> The any-time constraints can be violated if the Lipschitz constants and perturbation parameters are lower estimated or the minimum cost value is higher estimated.  Thus, knowing the upper bounds of the Lipschitz constants and telescoping parameters and the lower bound of the minimum cost value is conservative enough to guarantee the constraints by our algorithm. However, our analysis shows that the reward regret bound is sensitive to the errors of the parameters. As is shown in Theorem 5.1, using larger Lipschitz constants or perturbation function or using smaller minimum cost value in ACD can cause larger reward gap between the optimal ACD policy and the optimal unconstrained policy, resulting in a larger bound of the regret in Eqn. (2). This is because smaller safe action sets are obtained under this case, causing less flexibility to optimize the reward.
>
> `How can ACRL be extended to handle multiple costs or multiple constraints?`\
> If we can find a policy prior that can achieve satisfactory performance for all the cost metrics, we can define the non-empty safe action set as the intersection of multiple safe action sets in (7) corresponding to different cost metrics. In this way, ACD can always give feasible actions. The extension is challenging if multiple policy priors are used for different cost metrics and they are not consistent in action selection. It would be interesting to explore the methods to handle this case.
>
> **References**\
> [1] Ayoub, Alex, et al. "Model-based reinforcement learning with value-targeted regression." ICML, 2020.\
> [2] Zhou, Dongruo, Jiafan He, and Quanquan Gu. "Provably efficient reinforcement learning for discounted mdps with feature mapping." ICML, 2021.\
> [3] Yang, Zhuoran, et al. "On function approximation in reinforcement learning: Optimism in the face of large state spaces." NeurIPS 2020.\
> [4] Tennenholtz, Guy, et al. "Reinforcement Learning with History-Dependent Dynamic Contexts." ICML 2023. \
> [5] Agrawal, Akshay, et al. "Differentiable convex optimization layers." NeurIPS 2019.

---

> > ### Author Response · Authors · 2023-08-19
> >
> > We sincerely appreciate your time and effort in reviewing our paper. We hope that our responses have satisfactorily addressed your concerns, and we are eagerly open to continuing discussions.

---

> > ### Comment · Reviewer_skPV · 2023-08-20
> > **Thanks for the response**
> >
> > I want to thank the authors for their response. I find it helpful to address my questions. I would like to keep my original scores at this point.

---

> > > ### Author Response · Authors · 2023-08-20
> > > **Thanks for the response!**
> > >
> > > We thank the reviewer for reading our rebuttal and acknowledging the novelty of our work. We are more than happy to engage in any additional discussions that may be required.

---

### Official Review · Reviewer_SitE · 2023-07-06

**Soundness:** 3 good
**Presentation:** 2 fair
**Contribution:** 3 good
**Rating:** 6
**Confidence:** 3

**Summary:**

This paper proposed a novel problem called Anytime-constrained Markov Decision Process. Unlike CMDP, at each step, a constraint should be satisfied. A projection-based algorithm is proposed to solve the Anytime-constrained MDP. The authors gave a perturbation-based regret analysis for the algorithm.

**Strengths:**

The paper has great novelty. The anytime constrained MDP is a new problem formulation and has good value for real-world safety-critical problems. The perturbation-based analysis is solid to me.

**Weaknesses:**

Although the problem formulation is novel, the analysis might be weak. The assumptions of known Lipschitz constant, a prior policy, might be too strict. Thus, the regret analysis might have limited guidance for the real-world problem. I have seen that the authors already mentioned it in the limitations.

The writing needs to be improved. Some words in the important part of the paper are confusing. For example,

1. in Definition 3.3, the initialized and corresponding look quite confusing to me. Initial distribution usually refers to the states at step, or round, 0. The correspondence is also not clear. You might need to add more explanation to lines 153-159.

2. in line 328, the linear kernel assumption is mentioned inline, which is critical to the major contribution of this paper. The assumptions should be highlighted and discussed more.

**Questions:**

1. From intuitions of the constrained optimization theory, the projection-based algorithm might not be guaranteed to be optimal with non-convex objectives/constraints. Would you mind giving me some intuitions about why the projection-based algorithm will result in an asymptotically optimal ACD policy?



**Limitations:**

The authors addressed the limitations well.

---

> ### Author Rebuttal · Authors · 2023-08-03
>
> We appreciate the comments of the reviewer and address them as below.
>
> `Are the assumptions too strict?`\
> As discussed in limitations, the any-time constraint is significantly stricter than the expected constraints or the constraints with high probability, so it cannot be guaranteed without further assumptions. The assumptions to guarantee the any-time constraint include the known cost lower bound, the Lipschitz cost and transition functions with known Lipschitz constants, and the existence of a policy prior with telescoping property.
>
> The cost lower bound can be as small as zero, so this assumption is not strict. The Lipschitz cost functions are assumed in other MDPs [9]. The Lipschitz transition functions are common in sequential decision-making problems [3,4,10]. In mission-critical applications such as voltage control [6] and cooling control[7,8], the Lipschitz constants can be estimated based on prior knowledge. Additionally, the telescoping property holds for many policy priors, e.g. stable policies [5]. It is also proved for model predictive control [4], and it is assumed for other perturbation analysis [3].
>
> `Explanation of Definition 3.3 (Telescoping policy).`\
> Telescoping policy means when a policy is applied, if the state $x_{h_1}$ at round $h_1$ is perturbed to state $x_{h_1}'$, then the state $x_{h_2}$ at round $h_2$ is perturbed to $x_{h_2}'$, and the perturbation error $\|x_{h_2}-x_{h_2}'\|$ is bounded by $p(h_2-h_1)\|x_{h_1}-x_{h_1}'\|$.
>
> `The linear kernel assumption.`\
> The pseudo regret bound in Theorem 5.2 is a general bound that does not rely on the linear kernel assumption, but the linear kernel assumption is needed to get a regret bound with sub-linear property.  The regret bound in (16) relies on the linear kernel assumption. The linear kernel assumption is widely considered in regret analysis of RL [1,2].  We will highlight this assumption required for (16) and discuss more about the limitations.
>
> `Why does the projection-based algorithm result in an asymptotically optimal ACD policy?`\
> ACRL learns the ML model $\tilde{\pi}$ in Algorithm 1 to optimize expected reward for the new MDP environment defined by Algorithm 1. The optimal ACD policy $\pi^{\circ}$ is ACD with the optimal ML model $\tilde{\pi}^*$. However, please note that the optimal ACD policy does not necessarily gives the optimal expected reward for the (non-convex) constrained optimization in (1). This is because the safe action set (7) used in ACD is sufficient to guarantee the constraint in (1), but is not a necessary condition. That being said, the reward gap between the optimal ACD policy and the optimal policy of A-CMDP is upper bounded by the reward gap in Theorem 5.1 where the optimal ACD policy is compared with a stronger oracle - the optimal unconstrained policy. The reward gap shows that with larger $\lambda$ and $b$ (more relaxed constraint), the reward of the optimal ACD policy get closer to that of the optimal unconstrained policy.
>
>
> **References**\
> [1] Ayoub, Alex, et al. "Model-based reinforcement learning with value-targeted regression." ICML, 2020.\
> [2] Zhou, Dongruo, Jiafan He, and Quanquan Gu. "Provably efficient reinforcement learning for discounted mdps with feature mapping." ICML, 2021.\
> [3] Lin, Yiheng, et al. "Bounded-Regret MPC via Perturbation Analysis: Prediction Error, Constraints, and Nonlinearity." NeurIPS 2022.\
> [4] Lin, Yiheng, et al. "Perturbation-based regret analysis of predictive control in linear time varying systems." NeurIPS 2021.\
> [5] Tsukamoto, Hiroyasu, Soon-Jo Chung, and Jean-Jaques E. Slotine. "Contraction theory for nonlinear stability analysis and learning-based control: A tutorial overview." Annual Reviews in Control 52 (2021): 135-169.\
> [6] Shi, Yuanyuan, et al. "Stability constrained reinforcement learning for real-time voltage control." ACC 2022.\
> [7] Google Deepmind, Safety-first ai for autonomous data centre cooling and industrial control.\
> [8] Luo, Jerry, et al. "Controlling commercial cooling systems using reinforcement learning." arXiv preprint arXiv:2211.07357 (2022).\
> [9] Luo, Fan-Ming, et al. "A survey on model-based reinforcement learning." arXiv preprint arXiv:2206.09328 (2022). \
> [10] Gradu, Paula, John Hallman, and Elad Hazan. "Non-stochastic control with bandit feedback." NeurIPS 2020.

---

> > ### Author Response · Authors · 2023-08-19
> >
> > We sincerely appreciate the time and effort you dedicated to reviewing our paper. We hope that our responses have satisfactorily addressed your concerns, and we are enthusiastically open to further discussions.

---

### Decision · Program_Chairs · 2023-09-21

**Decision:**

Accept (poster)

**Comment:**

The paper introduces a stronger notion of constrained MDPs, where the cost constraint is imposed not in expectation but for any realization of the randomness. While the paper is borderline, overall the setting is well motivated and the theoretical/algorithmic contribution is interesting. For this reason I propose acceptance.

Nonetheless, I would like to encourage the authors to revise the paper by addressing the reviewers concerns and by integrating the rebuttal. In particular, I suggest to revise the following aspets:
- The assumptions seem overall reasonable. Nonetheless they seem quite strong and the lack of discussion on where they are used, and, if not a full lower bound, it would be useful to have a stronger conjecture on whether they are needed or how they could be relaxed.
- Expand the discussion on related work, in particular following the suggestions of Rev. MVJu.
- I would also encourage the authors to consider changing the title of the paper. "Anytime" is somehow overloaded wrt to the way it is used in bandit/online learning literature.